# Approaching Deep Learning through the Spectral Dynamics of Weights

## Abstract

We study the spectral dynamics of weights–the behavior of singular values and vectors during optimization–showing that they clarify and link many phenomena in deep learning. Through extensive experiments, covering small-scale "grokking" to large-scale tasks like image classification with ConvNets, image generation with UNets, speech recognition with LSTMs, and language modeling with Transformers, we identify a consistent bias with three key ingredients. First, singular values evolve unequally leading to rank minimization. As a result, top singular vectors stabilize well before the end of training, and lastly this happens without displaying alignment between neighboring layers used in several theoretical results. We show how this bias tracks the transition to generalization in grokking. We demonstrate more generally that weight decay enhances rank minimization beyond its role as a norm regularizer in practical systems. Moreover, we show that these spectral dynamics distinguish random label training from true labels, offering a novel perspective on this longstanding conundrum. Additionally, these dynamics reveal structure in well-performing sparse subnetworks (lottery tickets) and the shape of the loss surface through linear mode connectivity. Our findings suggest that spectral dynamics provide a coherent view that links the behavior of neural networks across diverse settings.

## 1 Introduction

Use of neural networks has exploded in the past decade. Capabilities are rapidly improving, and deployment is ever-increasing. Yet, although issues with these technologies now have social repercussions (Bender et al., 2021; Bommasani et al., 2021), many fundamental questions remain unanswered.

For instance, despite extensive research, we still lack a complete understanding of the implicit biases of neural networks trained via stochastic optimization (Neyshabur et al., 2014). Even basic questions on the role of regularization like weight decay (Hanson & Pratt, 1988; Krogh & Hertz, 1991; Zhang et al., 2018a) have only partial answers (Van Laarhoven, 2017; Andriushchenko et al., 2023; Yaras et al., 2023b). Perhaps most vexing, we lack a complete explanation for how neural networks generalize, despite having the capacity to perfectly memorize the training data (Zhang et al., 2021). Such an explanation may allow us to design better algorithms, however a lack of understanding makes the deployment of neural networks vulnerable to uninterpretable errors (Szegedy et al., 2013; Ilyas et al., 2019; Hendrycks et al., 2021; Zou et al., 2023).

Although theoretical explanations have been presented, these studies are often limited to special settings like deep linear networks (Arora et al., 2018; 2019) or infinite-width systems (Jacot et al., 2018), and arguments rely on unsubstantiated or impractical assumptions like near-zero initialization. On the empirical side, a growing body of work in interpretability has attempted to reverse-engineer neural networks (Rahaman et al., 2019; Barak et al., 2022; Nanda et al., 2023), but given the difficulty of the task, researchers have started at small scale and the methodology for analysis is quite bespoke and challenging to scale. A third category of work aims to understand empirical behavior on larger networks (Zhang et al., 2021; Huh et al., 2022; Yu & Wu, 2023), and compromises by focusing on more abstract objects like the gram matrix (Huh et al., 2022) or the Neural tangent kernel (NTK) (Fort et al., 2020). Thus the connection between the results of this category and the previous two is often hard to make.

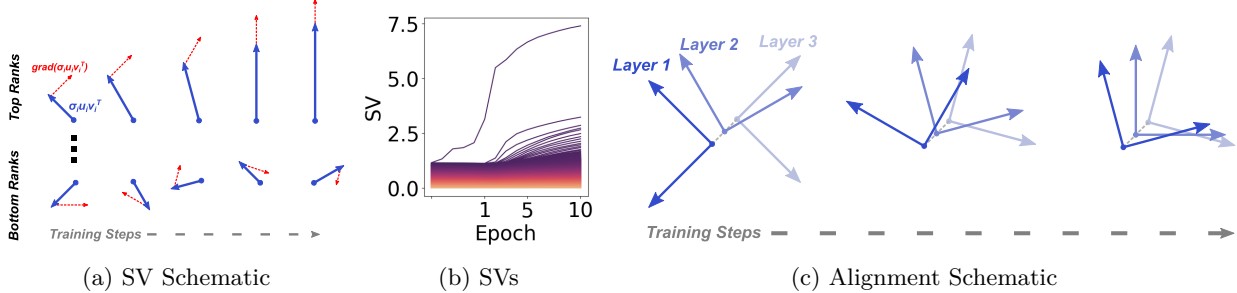

(a) SV Schematic  (b) SVs  (c) Alignment Schematic

Figure 1: (a) Schematic for the spectral dynamics of a weight matrix. As training proceeds, top singular vectors become stable and top singular values grow disproportionately large. (b) Singular value evolution for a single intermediate matrix parameter in a Transformer, where each line is a single singular value. Darker colors correspond to larger singular values while lighter ones corresponds to smaller values. We see a disproportionate trend where large singular values grow larger faster. (c) Previous works use SV basis alignment between layers to prove similar theoretical results, however actual alignment between consecutive layers is weak, which we describe in Section 5. We explore these spectral dynamics of weights and connect them to generalization, regularization, and seemingly unrelated phenomena like linear mode connectivity.

To bridge these gaps, we find a task- and architecture-agnostic view to link many disparate phenomena in deep learning, simultaneously studying image classification with ConvNets, image generation with UNets, speech recognition with LSTMs and language modeling with Transformers. Through extensive experiments, we examine the dynamics (i.e., evolution over training) of singular values and singular vectors of weight matrices—the spectral dynamics of weights. We observe a few key properties: singular values evolve unequally, with larger ones evolving faster. As a result, top singular vectors stabilize much faster. Finally, for top ranks we see alignment between neighboring layers' singular vectors, though this varies somewhat with the setting. We preview these phenomena in Figure 1. We are motivated to study these dynamics specifically as optimization is the fundamental process driving deep learning (Nagarajan & Kolter, 2019; Zhang et al., 2021), neural networks at their core are chained matrices and nonlinearities, and the SVD is fundamental to every matrix. The following paragraphs detail how these properties connect to generalization and many other phenomena.

**Sketch of the paper:** As a test bed for understanding generalization, Power et al. (2022) introduce the "grokking" phenomenon, where a small-scale model applied to arithmetic tasks initially achieves essentially zero training loss but performs poorly on validation data, then with much more training suddenly minimizes the validation loss (see Figure 2). In particular, Nanda et al. (2023) showed that in modular arithmetic, the feature learning that occurs during grokking could be reverse-engineered entirely from the final weight matrices. Although this description is precise, it is limited to arithmetic tasks. In Section 4, we notice a task-agnostic view of grokking, observing that the drop in validation loss during grokking coincides with the simultaneous discovery of low-rank solutions across all weight matrices in the network, whether it be modular arithmetic or image-classification. We also find that this transition relies on weight decay, echoing existing works (Lyu et al., 2023; Liu et al., 2023) as neither grokking nor low-rank weights occur without sufficient weight decay.

Though the common tie between low-rank weights and generalization in grokking suggests an intriguing explanation for generalization, grokking is typically studied on synthetic tasks with very small-scale models like single-layer Transformers or small MLPs and requires very particular hyperparameter settings (Power et al., 2022; Nanda et al., 2023; Gromov, 2023; Kumar et al., 2023). If our perspective is to be useful, we should obtain similar results in larger systems. Thus, we turn to common empirical tasks drawn from the literature like image classification, image generation, speech recognition and language modeling as well as varied and larger networks like VGG (Simonyan & Zisserman, 2014), UNet (Ronneberger et al., 2015), LSTM (Hochreiter & Schmidhuber, 1997b) and multi-layer Transformers (Vaswani et al., 2017).

In Section 5, we demonstrate that spectral dynamics are biased toward effective rank minimization across various practical neural networks in complex settings. Although this behavior echoes theoretical predictions in the deep linear setting, we find that the behavior of networks disagrees with a common theoretical assumption about low-rank dynamics: the alignment of singular vectors in consecutive layers (Saxe et al., 2014; Arora et al., 2018; 2019; Milanesi et al., 2021). Thus, the rank minimization mechanism differs from what the theory describes. Our hyperparameters are drawn from existing literature, thus the trend toward rank minimization coincides with well-generalizing networks across settings, echoing the observations in grokking.

One particularly notable ingredient for grokking was an extreme level of weight decay. In Section 6, we empirically connect rank minimization to weight decay, showing that weight decay promotes rank minimization across architectures and tasks, echoing findings in Section 4. In addition, in some cases, weight decay also appears to promote singular vector alignment in consecutive weight matrices despite the non-linearities between layers. Although weight decay explicitly penalizes norm, studying spectral dynamics allows us to observe these effects on rank and alignment. Such effects may lead to better generalization bounds by understanding the reason weight decay is useful, as the most obvious norm-based explanations are insufficient (Andriushchenko et al., 2023).

To further probe the rank-generalization connection, we turn to the classic memorization experiments of Zhang et al. (2021), who demonstrated that even small networks can memorize random labels, thus any arguments about generalization cannot be capacity-based alone. In Section 7 and Appendix A, we show that training with random labels leads to high-rank solutions, while rank with true labels is much lower. We also find that while random labels do not align consecutive layers, true labels do, which is surprising given the non-linearities between layers. This echoes prior discussion on rank and generalization. Through spectral dynamics, we see a clear distinction between generalizing and memorizing networks, which provides a foothold toward better theoretical understanding.

Our results suggest that viewing neural networks through the lens of spectral dynamics can shed light on several generalization-related phenomena, but we suspect there are broader connections. In the literature, many curious and unexplained phenomena regarding neural networks exist. We take two as case studies. First, the lottery ticket hypothesis (LTH) (Frankle & Carbin, 2018) and second, linear mode connectivity (LMC) (Nagarajan & Kolter, 2019; Frankle et al., 2020; Neyshabur et al., 2020). We find that global magnitude pruning, a standard procedure for finding lottery tickets, preserves top singular vectors and acts like low-rank pruning. We also see that the ability to interpolate between models in LMC strongly correlates with sharing top singular vectors. With these results, we note that the two phenomena can be seen as aspects of the spectral dynamics of weights, bringing them under the umbrella of prior sections. For detailed discussions, see Section 7 and Appendix B.

To summarize the discussion above, by studying the spectral dynamics of weights, we find:

- Grokking is intimately linked to rank minimization (Section 4);

- Rank minimization is a general phenomenon in more complex tasks and architectures (Section 5);

- Weight decay, norm regularization, enhances the rank minimization behavior (Section 6);

- Random-label training yields high-rank, unaligned parameters compared to true-label training (Section 7; and

- Top singular vectors are preserved when performing magnitude pruning and while linearly interpolating between connected modes (Section 7).

All of these phenomena and effects have previously been studied in isolation to varying degrees. By approaching deep learning through spectral dynamics, we aim to inform existing small-scale theoretical and experimental results, and link them to a much broader literature. Our hope is that this will provide a deeper footing for further theory and practice. Code for all experiments will be released.

## 2 Related Work

### 2.1 Singular Value Dynamics

Prior work on deep linear networks (Arora et al., 2019; Milanesi et al., 2021) suggests that rank minimization may better describe implicit regularization in deep matrix factorization than simple matrix norms. See Arora et al. (2018) (Appendix A) for a detailed argument. However, a critical assumption in these works is "balanced initialization." This means that for consecutive matrices $W_i$ and $W_{i+1}$ in the product matrix $\prod_j W_j$, we have $W_{i+1}^\top W_{i+1} = W_i W_i^\top$ at initialization. Decomposing these matrices with SVDs and leveraging orthogonality leads to matching left and right singular vectors between consecutive matrices. See Appendix C for a detailed explanation. Consequently, the product of the diagonals will evolve in a closed-form manner, with larger singular values growing faster than smaller ones. As shown by Arora et al. (2019), this translates to rank-minimizing behavior with increasing depth in the matrix products. This formula is also empirically validated for linear matrix factorization problems. Similar results have been derived for tensor products and other structured settings (Saxe et al., 2014; Yaras et al., 2023a). (Ji & Telgarsky, 2019) show that alignment between layers will happen specifically for deep linear networks with infinite training. Still, there is no reason to believe standard networks obey this balancedness condition under practical initialization procedures. In Section 5, we explore how these conclusions and assumptions hold for much larger, practical neural networks that are far from linear.

### 2.2 Low-Rank Properties

Another line of research focuses on more general low-rank biases. Early work explored norms as an implicit bias (Gunasekar et al., 2017). Theoretical analyses reveal that norms or closed-form functions of weights might be insufficient to explain implicit regularization, but they do not necessarily contradict the possibility of rank minimization (Razin & Cohen, 2020; Vardi & Shamir, 2021). Numerous studies investigate low-rank biases in various matrices, including the Jacobian (Pennington et al., 2018), weight matrices (Le & Jegelka, 2021; Martin & Mahoney, 2020; 2021; Frei et al., 2022; Ongie & Willett, 2022), Gram matrix (Huh et al., 2022), and features (Yu & Wu, 2023; Feng et al., 2022). Additionally, research suggests that dynamics influence the decay of rank (Li et al., 2020; Chen et al., 2023; Wang & Jacot, 2023). Orthogonally, weight decay has a long history as a regularizer explicitly penalizing parameter norm, which can be used for norm-based generalization bounds (Bartlett, 1996), but these bounds do not seem to explain the success of practical systems (Nagarajan & Kolter, 2019; Jiang et al., 2019). Some works establish connections between weight decay and rank minimization in idealized settings (Ziyin et al., 2022; Galanti et al., 2022; Zangrando et al., 2024; Ergen & Pilanci, 2023; Parhi & Nowak, 2023; Shenouda et al., 2023), which may be connected to generalization (Razin & Cohen, 2020). We are particularly interested in how far these connections extend beyond ideal settings to practice.

Though these prior results are interesting in their own right, our goal is to link them together on a much broader suite of experiments and show further unexplored consequences.

## 3 Common Quantities

Throughout the rest of the paper we will rely on a number of common quantities for our analysis. Inspired by work in the deep linear case (Saxe et al., 2014; Arora et al., 2019; Milanesi et al., 2021; Yaras et al., 2023b), we track the evolution of singular values for individual weight matrices. To gain a high-level overview of all matrix parameters, we compute the (normalized) effective rank of a matrix $W$ (Roy & Vetterli, 2007) with rank $R$ as

$$\text{EffRank}(W) := -\sum_{i=1}^{R} \frac{\sigma_i}{\sum_j \sigma_j} \log \frac{\sigma_i}{\sum_j \sigma_j} \ , \tag{1}$$

$$\text{NormEffRank}(W) := \frac{\text{EffRank}(W)}{R} \ , \tag{2}$$

where $\sigma_i$'s are the singular values of matrix $W$ and $\mathrm{EffRank}(W)$ is the entropy of the normalized singular value distribution. As the probability mass concentrates, the effective rank decreases. We plot $\mathrm{NormEffRank}(W)$ to compare across layers and time.

In addition, inspired by the assumptions of balancedness made by prior work (Arora et al., 2018; 2019), we examine the alignment of consecutive weight matrices in the Transformer. To examine and quantify this alignment between SVDs of consecutive matrices in a network at training time $t$, i.e.,

$$W_i = \sum_{j=1}^{R} \sigma_j(t) u_j(t) v_j(t)^\top, \qquad W_{i+1} = \sum_{k=1}^{R} \sigma'_k(t) u'_k(t) v'_k(t)^\top \ ,$$

we compute the inner products between neighboring singular vectors,

$$A(t)_{jk} = |\langle u_j(t), v'_k(t) \rangle| \ , \tag{3}$$

where the absolute value is taken to ignore sign flips in the SVD computation. We then plot the diagonal of this matrix $A(t)_{ii} \ \forall \ i \leq 100$ over time, which we focus on because we observed the most signal here. See Appendix F.4 for more details. These plots give us a sense as to whether simultaneous diagonalization occurs at least in the top ranks. For exact details on how alignment is computed for different architectures and layers more complex than the fully connected case, see Appendix D.

We not only employ the alignment matrix defined in Eqn. 3, but also derive and plot a scalar measure. We focus on the top diagonal entries as we observed they typically contained the most structure (see Figure 23 for an example):

$$a(t) = \frac{1}{10} \sum_{i=1}^{10} A(t)_{ii}. \tag{4}$$

For specific details on calculating this measure in diverse architectures and complex layers (beyond fully connected layers), please see Appendix D.

## 4 Grokking and Rank Minimization

Power et al. (2022) first noticed a surprising phenomenon they called "grokking" where models quickly fit the training data on toy tasks, then after a long period of training, very quickly generalize on the validation data (see Figure 2). Later, others found that this phenomenon can occur on a logarithmic timescale (Thilak et al., 2022) on simpler models and different datasets (Liu et al., 2022; Gromov, 2023; Kumar et al., 2023; Xu et al., 2023). In addition, weight decay seems to be a critical ingredient (Lyu et al., 2023; Liu et al., 2023; Tan & Huang, 2023).

Motivated by experimental results that show the importance of weight decay for grokking (Power et al., 2022; Lyu et al., 2023; Liu et al., 2023), and theoretical work that connects low-rank weights, generalization and weight decay (Razin & Cohen, 2020; Galanti et al., 2022; Timor et al., 2023; Yaras et al., 2023b; Zangrando et al., 2024), we evaluate the potential connection between parameter rank and grokking in neural networks. Low-rank weights would naturally complement other descriptions such as Fourier decomposition (Nanda et al., 2023), the simplification of linear decision boundaries (Humayun et al., 2024), the connection to double descent (Davies et al., 2022), and the discovery of a sparse solution (Merrill et al., 2023).

We replicate grokking in two settings: a single-layer Transformer for modular addition (Nanda et al., 2023), and a 12-layer MLP for MNIST image classification (Fan et al., 2024) (see Appendix D for details). We select these settings as they are somewhat more realistic than other tasks, require more complex architectures, and display grokking on a linear time scale.

In Figure 2, we see a tight connection: the sudden drop in validation loss coincides precisely with the onset of low-rank behavior in the singular values. Examining inter-layer alignment during training, we observe that the final low-rank solution gradually emerges from the model's middle ranks. As we show later, in the absence of weight decay no low-rank solution seems to develop (Figure 19). Additionally, when using 90% of the data and no weight decay, generalization still coincides with effective rank minimization. Fan et al. (2024)

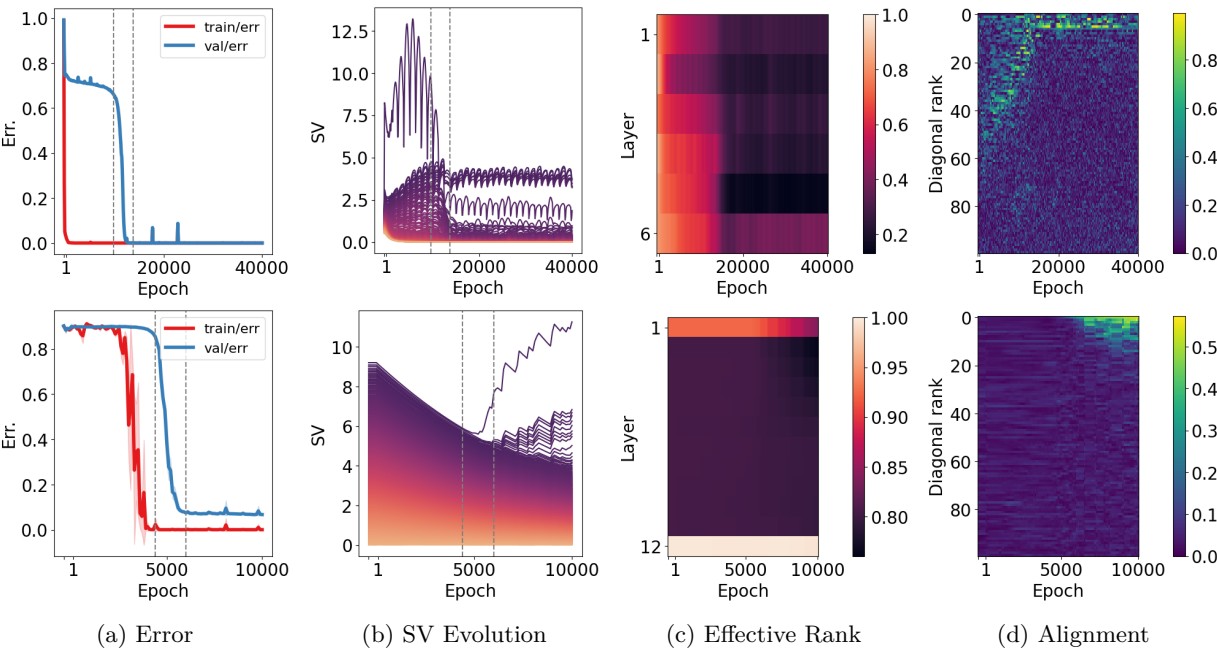

(a) Error  (b) SV Evolution  (c) Effective Rank  (d) Alignment

Figure 2: **Grokking and Spectral Dynamics. Top row:** Grokking for Transformers on modular addition (Nanda et al., 2023) for a single seed due to different convergence times. See Figure 22 for averaging over multiple seeds. **Bottom row:** Grokking for a 12-layer MLP on MNIST (Fan et al., 2024). **1st column:** Training and validation error. Shaded regions correspond to standard deviation, where large deviations are due to different seeds converging at different times. **2nd column:** A visualization of singular value evolution for the first attention parameter and the second MLP layer, where each line represents a single singular value and the color represents the rank. **3rd column:** Effective rank of all layers (matrix parameters) over time (Eqn. 1). **4th column:** A visualization of the alignment (Eqn. 3) between the embedding and the first attention parameter, and the first and second MLP layers, where the $y$-axis corresponds to index $i$ of the diagonal. We see that grokking co-occurs with a transition to low-rank weights. In addition, there is an alignment that begins early in training that evolves up the diagonal. In the image classification case, we see a similar rank transition, though alignment does not evolve up the diagonal (see Figure 23).

noted that in deep MLPs, grokking coincided with a feature rank decrease, which agrees with the parameter rank decrease that we see here. The familiar reader will also note that Nanda et al. (2023) previously showed that the particular solution found in modular addition is a low-rank Fourier decomposition, so our observations on low-rank weights will follow, yet the same structure also applies to the MLP where such reverse-engineering is difficult. In the following sections, we argue that rank minimization is a perspective that can be applied in more complex settings when one does not know what to look for in the weights. It may thus be possible to interpret the neural network via the top ranks (Praggastis et al., 2022). Specifically, we will show in Section 6 that large amounts of weight decay have a strong rank-regularizing effect, so one way to understand grokking is that the network first memorizes the training data, and with further training the rank regularization of weight decay pushes toward generalization.

## 5  Spectral Dynamics Across Tasks

Inspired by the results on grokking and prior work on deep linear networks that studies the evolution of the SVD of the weight matrices (Saxe et al., 2014; Arora et al., 2018; 2019; Milanesi et al., 2021; Yaras et al., 2023a), we apply the same analysis to larger, more practical systems. We show that the trends we saw in the analysis of grokking mostly hold true across networks and tasks at a much larger scale, though our findings do deviate from theoretical settings.

### 5.1 Methodology

Our experiments aim to examine reasonably sized neural networks across a variety of tasks. We select models and tasks that are representative of current applications. Specifically, we consider:

- **Image classification** with CNNs (VGG-16 (Simonyan & Zisserman, 2014)) on CIFAR10 (Krizhevsky, 2009);
- **Image generation** through diffusion with UNets (Ronneberger et al., 2015) on MNIST (LeCun, 1998);
- **Speech recognition** with LSTMs (Hochreiter & Schmidhuber, 1997b) on LibriSpeech (Panayotov et al., 2015); and
- **Language modeling** with Transformers (Vaswani et al., 2017) on Wikitext-103 (Merity et al., 2016).

Training hundreds of runs for each of the settings above is computationally expensive, limiting the scale of models we can explore. We primarily adopt hyperparameters from existing literature, with minor modifications for simplicity. This ensures that any correlations observed are likely a reflection of common practices, and not bias introduced on our part. We also provide evidence with larger language models (up to 3B parameters) in Appendix D.5, from the Pythia suite (Biderman et al., 2023).

The primary evidence in this section comes from computing the SVDs of weight matrices within the models; we disregard 1D bias and normalization parameters in our analysis. Previous research suggests that these parameters are not always crucial for performance (Zhang et al., 2018b; Mohan et al., 2019; Karras et al., 2023), and many large models do not use them (Raffel et al., 2020; Grattafiori et al., 2024). As the matrices are the vast majority of the parameters, we believe they are the primary object of interest and it is worthwhile focusing on them. Due to the large number of matrices in these models, we present plots of individual layers' matrix parameters and statistics that summarize behavior across layers for conciseness. We generated hundreds of thousands of plots were as part of this study, making it impossible to include them all. Full experimental details, including the choice of hyperparameters, are available in Appendix D.

### 5.2 Effective Rank Minimization

Building on theoretical (Saxe et al., 2014; Arora et al., 2019; Milanesi et al., 2021; Boix-Adserà et al., 2023; Yaras et al., 2023a) and empirical (Dittmer et al., 2019; Martin & Mahoney, 2020; 2021; Boix-Adserà et al., 2023) findings, we investigate effective rank minimization across larger models and a more diverse array of tasks. Figure 3 reveals a consistent trend: the effective rank of network parameters generally decreases throughout training, regardless of the specific parameter or network architecture. This suggests a progressive "simplification" of the network as training progresses.

We further conduct a singular-value pruning experiment to explore the relationship between low-rank behavior and model performance. We prune either the top or bottom half of the singular values for each weight matrix in the network and then evaluate the pruned model at each training step. Given their importance in $\mathcal{L}^2$ space, we expect the top singular values to capture the information most critical to the network's function. Figure 4 confirms this, demonstrating that the pruned parameters, without further training, can closely approximate the full model's performance. It is not necessarily obvious that pruning would have this effect. In particular, simultaneously pruning lower components across all layers may lead to losing some critical signal that must be passed between layers, or it could be that small-magnitude singular values may provide some important regularizing noise. This result is reminiscent of prior work that uses low-rank approximations for efficiency or performance (Yu et al., 2017; Sharma et al., 2023; Chen et al., 2024), or work that explicitly optimizes for low-rank networks (Wang et al., 2021; Schotthöfer et al., 2022). Here we point out that the reason this is possible is due to the dynamics of the singular values themselves. In later sections, we will rely on this observation that large singular values are more critical to the function of the network.

### 5.3 Alignment of Singular Vectors Between Layers

Similar to the analysis of grokking, we investigate the alignment between consecutive layers in the larger neural networks considered in this section.

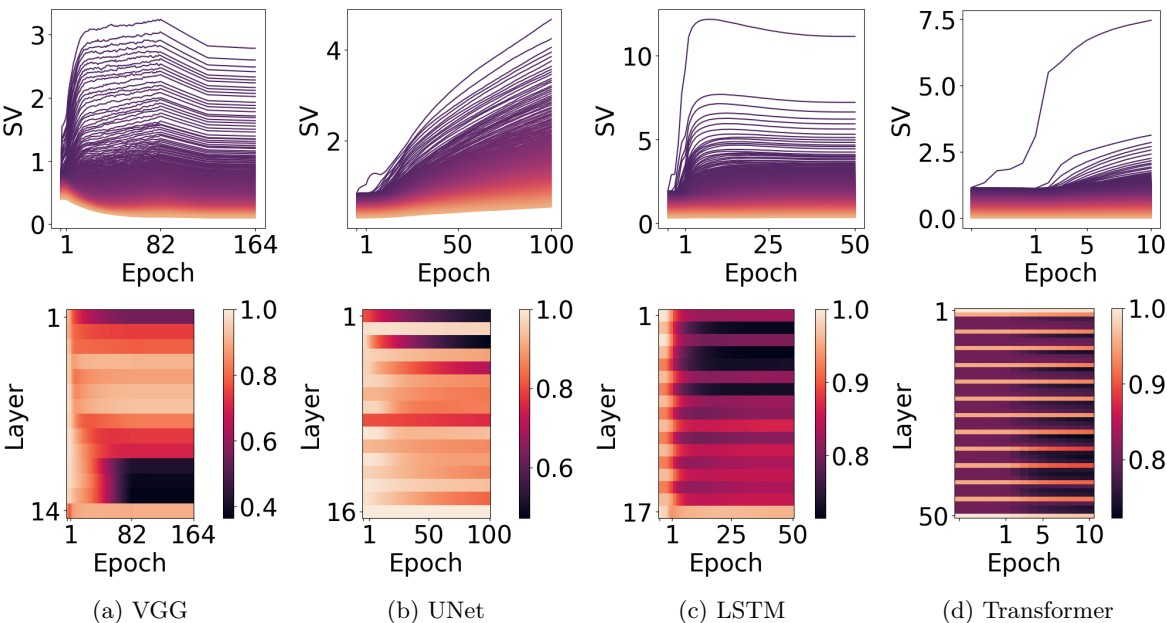

(a) VGG       (b) UNet       (c) LSTM       (d) Transformer

Figure 3: **Top row:** Singular value evolution for a single matrix in the middle of each model. Each line represents a singular value, darker colors are larger values, while lighter colors are smaller values. Notice the unequal evolution where top singular values grow at a disproportionate rate. **Bottom row:** Normalized effective rank (Eqn. 1) evolution visualized in color for all matrix parameters across architectures and time. As we move down the $y$-axis, the depth of the parameters in the model increases, while the $x$-axis tracks training time. The axis label "Layer" is shorthand for "matrix parameter", e.g. in the Transformer we visualize each of the $W_q, W_k, W_v, MLP_1, MLP_2$ parameters for each block. Notice decreasing effective rank with training time across nearly all parameters, though the magnitude differs across layers (indicated by absolute color). The block-like patterns for VGG are likely due to different channel dimension sizes. The banding in the UNet, LSTM, and Transformer is due to the differences between convolutional and linear layers, residual block connections, and attention and fully connected layers, respectively. The sharp transitions through training in VGG are due to $10\times$ learning rate decays.

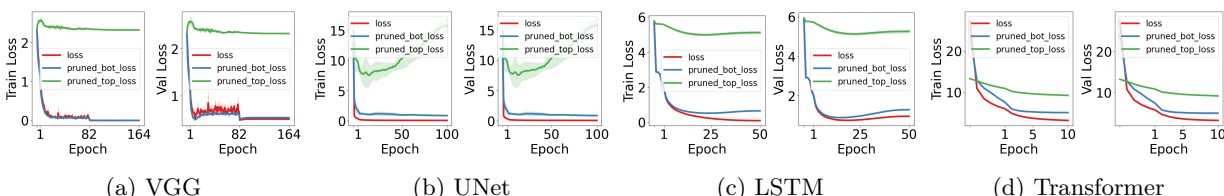

(a) VGG       (b) UNet       (c) LSTM       (d) Transformer

Figure 4: **Left plot:** Training loss. **Right plot:** Validation loss. Red is the full model. Blue is post-training pruning the bottom half of the SVD for every matrix in the model that is not the final layer. Green is post-training pruning the top half of the SVD. Notice that for all models, keeping the top half of the SVD is close to the full model performance, supporting the idea that the top directions provide a good approximation to the function. We specifically avoid pruning the final layer as for some tasks it is so low-rank that pruning further affects performance in an anomalous way (Frankle et al., 2020).

Figure 5 reveals a key finding: the theoretical assumption of **balanced initialization**, which posits aligned singular value decompositions (SVDs) between weight matrices (Arora et al., 2018; Saxe et al., 2014), does not hold true at the start of training in these larger networks. Additionally, unlike in the linear case discussed in Du et al. (2018), the alignment does not appear to remain static throughout training. However, a weak signal of alignment in the top ranks develops and disappears. We are very far from the theoretical settings of prior work (Du et al., 2018; Arora et al., 2019; Mulayoff & Michaeli, 2020) that use balanced or near-zero

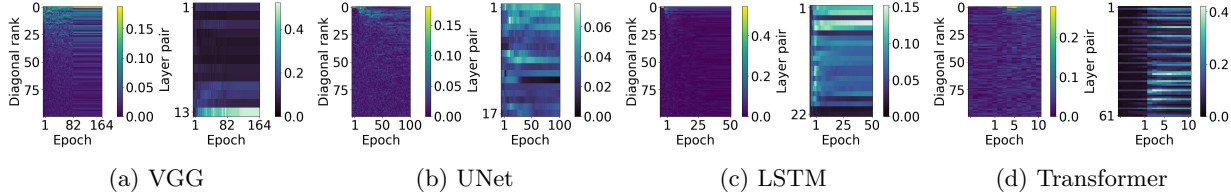

| (a) VGG | (b) UNet | (c) LSTM | (d) Transformer |

Figure 5: Neighboring layer alignment of singular vectors. **Left plot:** The diagonal of the alignment matrix $A(t)_{ii}$ (Eqn. 3) vs. training time for a single pair of matrices in the middle of each model. We see a small amount of alignment in the top ranks between layers shortly after training begins, but this becomes more diffuse over time. **Right plot:** Alignment metric (Eqn. 4) for pairs of matrices for depth vs. training time. It is hard to make out a global trend across models, though the LSTM shows a weak signal around Epoch 1 when the initial alignment occurs, and the Transformer case has a banding pattern with depth due to alignment between the query and key matrices that have no nonlinearity in between.

initialization, and the strange trends mean that existing theoretical models do not capture the complexities of neural network training.

## 6 The Effect of Weight Decay

In light of the previously observed evolution of singular values, we investigate a proposed effect of weight decay. Though weight decay explicitly penalizes the norm of weights, there is evidence that complicates the connection between the norm and generalization for neural networks (Razin & Cohen, 2020; Andriushchenko et al., 2023), meaning that we do not have a full understanding as to why weight decay may be useful. Alternatively, some theoretical (Boix-Adserà et al., 2023; Razin & Cohen, 2020; Yaras et al., 2023a; Timor et al., 2023; Ongie & Willett, 2022; Galanti et al., 2022; Zangrando et al., 2024) and empirical works (Galanti et al., 2022; Boix-Adserà et al., 2023) propose a connection between weight decay, generalization and the rank of matrices in constrained settings. Still, comprehensive evidence for larger empirical networks is missing.

We speculate on the intuition behind the mechanism in more practical settings. In its simplest form, weight decay involves the optimization: $\arg\min_W \mathcal{L}(W) + \lambda\|W\|_F^2$, where $\|W\|_F^2 = \sum_{i=1}^R \sigma_i^2$, where $\sigma_i$ are the singular values of weight matrix $W$ with rank $R$. We saw previously that larger singular values of neural networks grow faster (Fig. 3, top row) and that the top singular vectors are much more useful for minimizing task loss than the bottom ones (Fig. 4). Thus, with minor weight decay regularization, one straightforward solution for the network may be to minimize the rank of a given weight matrix while preserving the top singular values to minimize $\mathcal{L}(W)$. Timor et al. (2023) argue a similar effect for simple systems: if all singular values are less than one, the norm of activations will shrink with depth, so as depth grows very large, any input signal will converge to zero, making it impossible to learn. Thus, it is better for a few singular values to be sufficiently large while the rest can be very small.

Figure 6 shows that adding weight decay produces this exact low-rank behavior, while too much weight decay leads to complete norm collapse. The exact choice of "too much" varies across architectures and tasks. Despite the low-rank regularization, we do not see particularly tight alignment (Eqn. 3) in the top singular vectors, except in the highest weight decay Transformer (see Figure 17 in the Appendix). The alignment in this case is reminiscent of the balancedness condition (Arora et al., 2018; 2019; Du et al., 2018), though the Transformer considered here has nonlinearities and a much more complex structure.

Orthogonally, we provide additional evidence in Appendix D, where Figure 18 shows that the solutions with moderate weight decay can perform better than without, while with very high weight decay models still generalize, even though they are much lower rank. It is difficult to argue such a simple trend as "lower rank equals better generalization" because one does not know the minimal rank necessary for a given task. Language modeling may require some form of memorization for long-tail words and thus a low-rank solution may be impossible, while 10-class image classification may not have this property. Still, we note that the role of weight decay in improving generalization is tied up with its function as a rank regularizer. In addition,

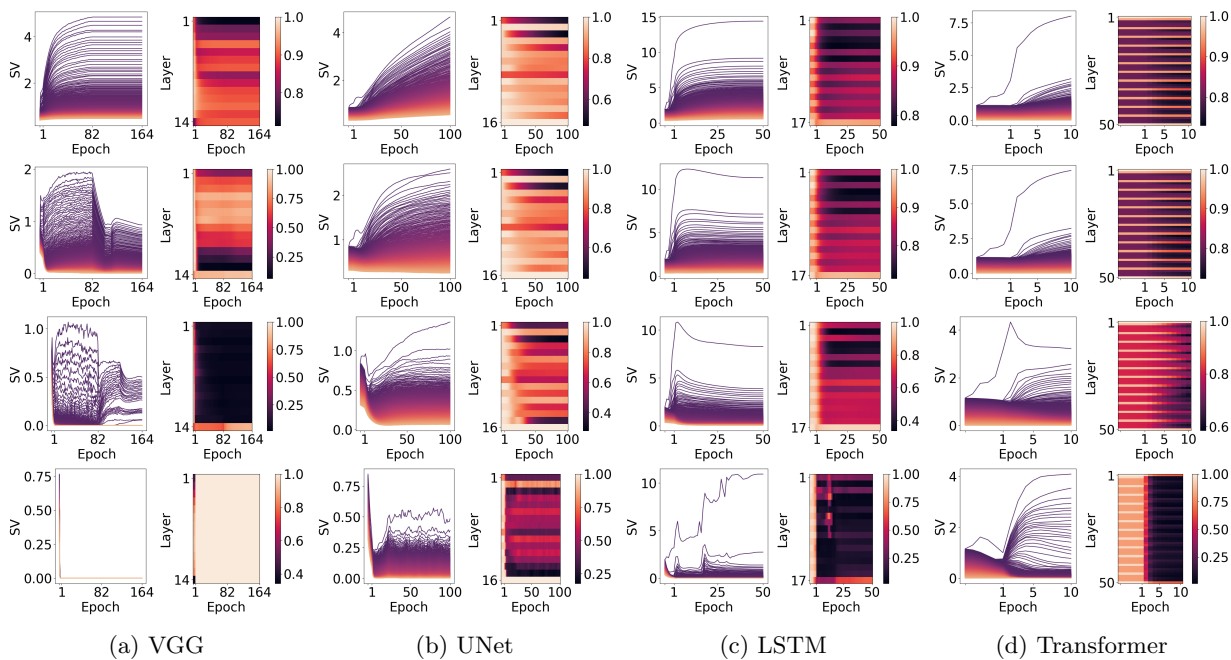

Figure 6: Singular value evolution for a single matrix and normalized effective rank (Eqn. 1) across matrices over time, where the rows use differing amounts of weight decay. From top to bottom, for VGG we use coefficients $\{0, 0.001, 0.01, 0.1\}$, while for other networks we use coefficients $\{0, 0.1, 1, 10\}$. Higher weight decay coefficients promote more aggressive rank minimization. VGG uses SGD with momentum, while the rest use AdamW (Loshchilov & Hutter, 2017), which may explain the earlier norm collapse. We provide an additional version with shared color bars in Figure 25.

although we lack precise tools to interpret complex models entirely, when there are only a few ranks per matrix, it may become possible to extend analysis efforts (Nanda et al., 2023; Praggastis et al., 2022) to more complex domains.

# 7 Additional Connections

Here we briefly preview some connections between spectral dynamics and additional phenomena.

**Memorization vs. Generalization:** In Figure 7, we replicate the core memorization experiment of Zhang et al. (2021), which highlighted the ability of modern neural networks to memorize even random labels perfectly. We find that when training with random labels, we obtain networks with higher-rank final parameters as opposed to when training with true labels. Thus, the spectral dynamics can distinguish between memorization (of random labels) and generalization. We also see an alignment (Eqn. 3) structure between the middle layers that disappears with random labels, perhaps as it is necessary in order for the network to pass signals from input to output. This echoes the pattern of grokking in Section 4, where generalization came with a transition to low-rank and alignment in middle layers. We expand this experiment to more settings in Appendix A with additional figures.

**Lottery Tickets:** On very small networks, Frankle & Carbin (2018) found the existence of sparse subnetworks via magnitude pruning, keeping only the top $p\%$ of weights globally by magnitude, that could train to similar performance as the full network. For larger image classification networks, Frankle et al. (2020) observed that in order to find such sparse subnetworks, it was necessary to train till the end in order to acquire the pruning mask that could be used retroactively in training. This curious observation still lacks a compelling explanation. We show that such global magnitude pruning functions similarly to low-rank pruning, thus the lottery ticket masks found by rewinding (Frankle et al., 2020) are effectively low-rank masks for the singular components that will become important at the end of training. Training the masked

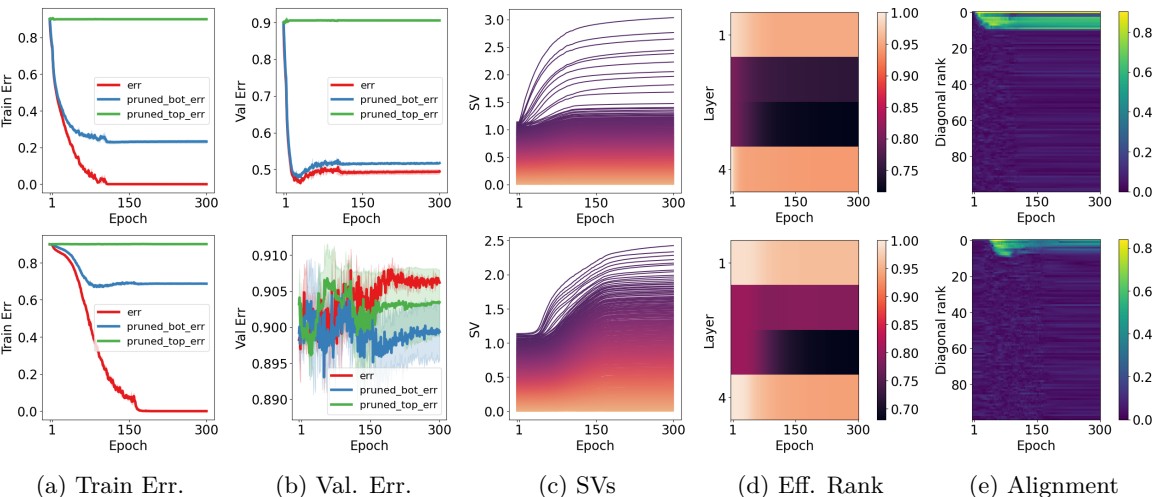

(a) Train Err.  (b) Val. Err.  (c) SVs  (d) Eff. Rank  (e) Alignment

Figure 7: **Top row:** results with true labels. **Bottom row:** results with random labels. We see that the middle layers have a lower effective rank (Eqn. 1) when using true labels and that alignment (Eqn. 3) in the middle layers persists throughout training, unlike in the random label case. We emphasize this alignment occurs despite the nonlinearities.

network leads to similar dynamics in these components. However, taking masks from too early in training leads to poor approximation of these final components and simultaneously stunts training. We provide detail on this discussion in Appendix B.

**Linear Mode Connectivity (LMC):** Linear Mode Connectivity (Nagarajan & Kolter, 2019; Frankle et al., 2020) refers to the property that models that share a portion of the training trajectory can be averaged in weight-space to yield a stronger model (Wortsman et al., 2022; Ramesh et al., 2022). This phenomenon indicates that, after some training, the loss surface is quite convex in a subspace, even though the optimization problem is theoretically highly non-convex. As prior work has found fine-tuning from pre-trained models stays in this convex space (Neyshabur et al., 2020; Li et al., 2022; Sadrtdinov et al., 2023), an explanation for what underlies LMC could help to clarify the role of pre-training, and may lead to faster fine-tuning. We show that LMC is tied with singular vector sharing. In particular, when models display LMC, they share top singular vectors between weights (see Figure 8), and when they do not they also do not share parameters (Appendix B.3). We argue this is an outcome of the early stability of top singular vectors, which arises due to the unequal evolution of singular values. It is also straightforward to explain the large Euclidean distance between checkpoints (Frankle et al., 2020; Yunis et al., 2022) that can be averaged, as they only share a very small portion of the parameter space. Thus, LMC, and by extension, model-averaging, is deeply intertwined with the dynamics of singular values that we explore in Section 5. The full discussion is deferred to Appendix B.

## 8  Discussion

We provide an empirical perspective to understand deep learning through SVD dynamics. We first note a tendency toward rank minimization on a small scale in grokking, then expand these findings to practical networks and tasks. In addition, we find that weight decay, though it explicitly penalizes norm, implicitly promotes this low-rank bias. We also show that training with random and true labels differ in the rank and alignment of solutions found by optimization, echoing the rank-generalization connection during grokking. We go beyond remarks on generalization and show that magnitude pruning for lottery tickets acts similarly to low-rank pruning, and LMC coincides with the sharing of top singular vectors between checkpoints.

A comprehensive theory for all these results remains elusive. Our goal in this work is to provide these observations as a platform for a deeper understanding of deep learning. Notably, the observed spectral dynamics

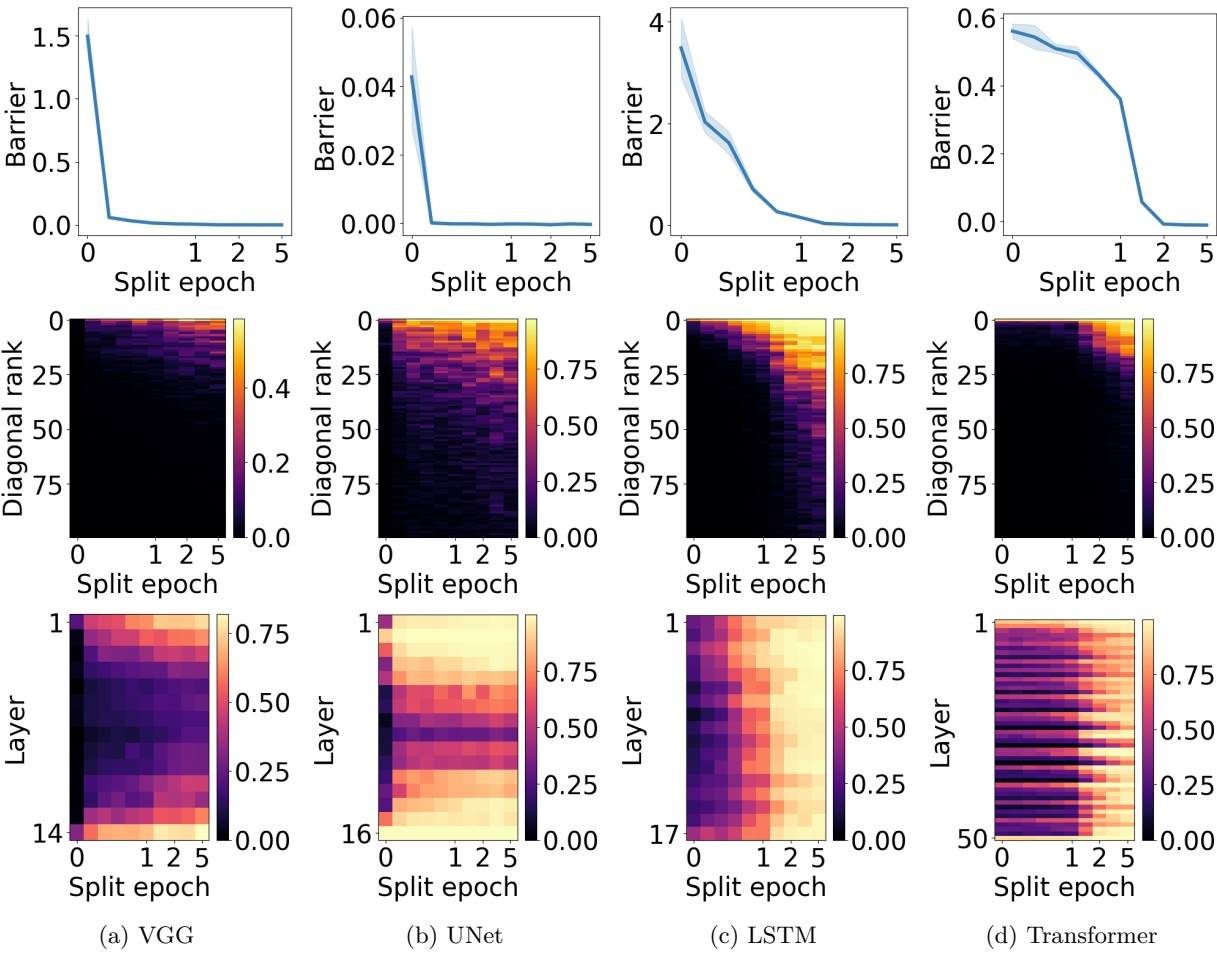

Figure 8: **Top row:** Barrier size (Eqn. 6) vs. split step. **Middle row:** singular vector agreement for a single matrix parameter between branch endpoints that share a common trunk. **Bottom row:** summary statistic for singular vector agreement across layers vs. split step. We see that as models exhibit LMC, they also share a small number of top singular vectors. We only show agreement for 100 dimensions as the rest are near zero.

appear consistent across diverse settings, even without restrictive assumptions like balanced initialization, linearity, or small weight scales. This suggests a common underlying mechanism that further theoretical work may be able to model. One limitation our study is the focus on linear alignment between neighboring layers. In linear systems, this alignment is a proxy for the network's ability to pass signals from input to output, but once nonlinearities are introduced, it is unclear how this proxy applies. We would guess there is still alignment in larger systems, but it may need to be viewed post-activation for particular examples. This is a promising direction for future exploration.

On the empirical side, several interesting problems present themselves. Interpretability of neural networks is a growing area of research (Nanda et al., 2023), and there already exist efforts to interpret singular vectors of convolutional weights (Praggastis et al., 2022). There may also be connections to other unexplained phenomena such as double descent (Belkin et al., 2019; Nakkiran et al., 2021; Davies et al., 2022) or adversarial examples (Szegedy et al., 2013; Ilyas et al., 2019; Hendrycks et al., 2021). For example, is it actually the case that low-rank models are more robust? The answers to these questions may help design better optimizers or diagnose deployment risks in the wild. There are also concerns of safety (Bai et al., 2022; Mazeika et al., 2024) that a better understanding of neural networks can alleviate (Burns et al., 2023; Park et al., 2024). In particular with low-rank models, interpretability may become easier as there is much less model

to study. We present this large empirical exploration in an effort to deepen scientific understanding, provide fertile ground for new theoretical assumptions, and offer inspiration for better engineering. We believe the developed perspective will be useful in bridging gaps between many different attempts to explain neural networks.

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

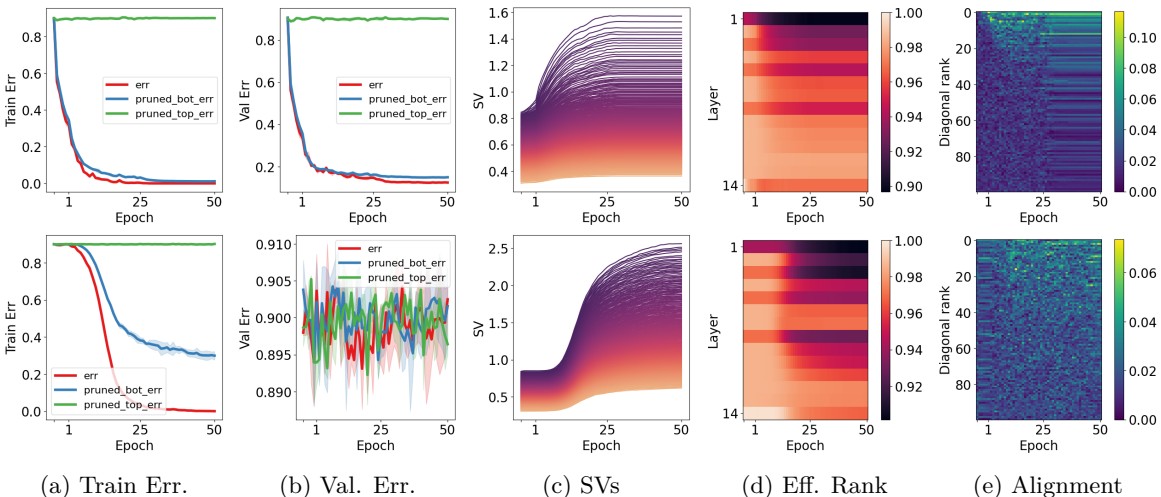

Figure 9: Dynamics with random labels for VGG. **Top row:** results with true labels. **Bottom row:** results with random labels. We see that the middle layers have a lower effective rank (Eqn. 1) when using true labels and that alignment (Eqn. 3) in the middle layers persists throughout training. The results are less stark in the VGG case, but similar to the MLP.

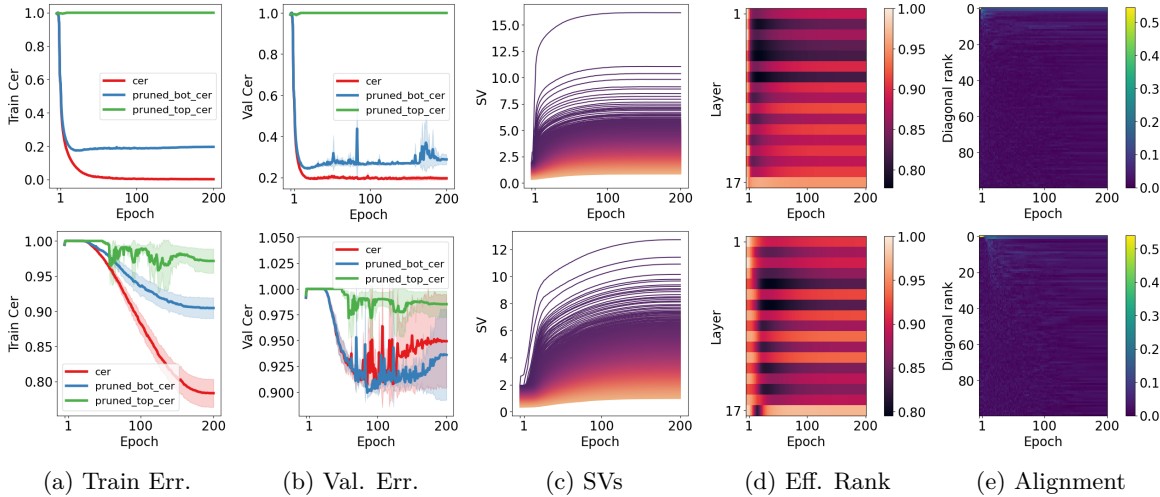

Figure 10: Dynamics with random labels for LSTM. **Top row:** results with true labels. **Bottom row:** results with random labels. We see that the middle layers have a lower effective rank (Eqn. 1) when using true labels and that alignment (Eqn. 3) in the middle layers persists throughout training. Though the LSTM doesn't fit the random labels perfectly, the results are qualitatively similar to the other cases.

## A   Spectral Dynamics with Random Labels

Given the observations connecting generalization and rank thus far, and the enlightening view on the implicit effects of weight decay, we are interested in seeing whether the perspective developed sheds any light on the classic random label memorization experiments of Zhang et al. (2021).

Similar to Zhang et al. (2021), we train a MLP, VGG and an LSTM to fit random or true labels. Please see Appendix D for the details regarding the experimental setup. Zhang et al. (2021) decay the learning rate to zero, and the random label experiments only converge late in training. Consequently, we use a constant learning rate to avoid confounding factors. We see in Figure 7 that both cases for the MLP are able to achieve zero error, though with different singular value evolution and alignment (Eqn. 3) in the middle layer.

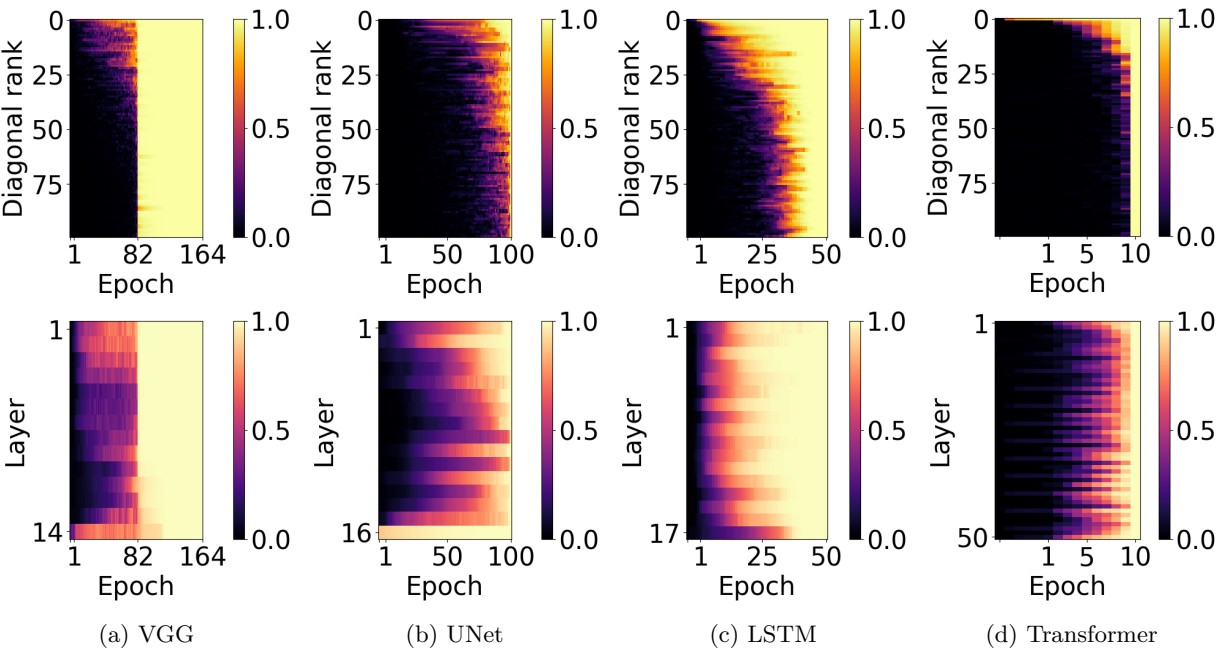

Figure 11: **Top row:** Singular vector agreement for a single matrix in the middle of each model (diagonal of Eqn. 5). Notice top singular vectors become stable in direction earlier. **Bottom row:** Summary score for each matrix across architectures. As we move down the $y$-axis, the depth of the parameters in the model increases, while the $x$-axis tracks training time. The sharp transition midway through training in the VGG case is likely due to a 10x learning rate decay.

Surprisingly in Figure 7, we see that with true labels the inner layers are low rank, while with random labels they are much higher rank. This may be explained by the shared structure in the true classes of the dataset, which manifests in the parameters. Even more surprisingly, we find here that even without weight decay, inner layers align with true labels, while with random labels, this alignment occurs and then disappears with more training. This is particularly intriguing as there are non-linearities that could theoretically separate the network from the linear case, and yet strong alignment occurs despite that. Such alignment has not yet been leveraged by existing theory, and might provide structured assumptions for new understanding. Results on the VGG (Figure 9) and LSTM (Figure 10) are qualitatively quite similar, though weaker in alignment. In summary, these results suggest that viewing generalization through the lens of rank and alignment may be fruitful, deepening the connection previously revealed.

## B  Beyond Generalization

We have seen over the course of many experiments that deep models are biased toward low rank, and that there is a tempting connection between rank minimization and generalization. Still, spectral dynamics may have broader connections. In the following subsections, we explore two unexplained phenomena: lottery tickets (Frankle & Carbin, 2018) and linear mode connectivity (Frankle et al., 2020). Beyond shedding further light on neural networks, these phenomena have implications for more efficient inference and storage, as well as understanding the importance of pretraining (Neyshabur et al., 2020). We find that lottery tickets are a sparse approximation of final-checkpoint top singular vectors. The ability to linearly interpolate between faraway checkpoints and improve performance coincides strongly with top singular vector sharing between checkpoints. Such observations may form a foundation for a better understanding compression and model averaging (Wortsman et al., 2022; Ilharco et al., 2022).

## B.1 Top Singular Vectors Become Stable Earlier

Before we explore the phenomena, we first make another observation that will be helpful. As top singular values grow disproportionately large, it would be natural that top singular vectors become stable in direction as the gradients remain small. To demonstrate this, for a given matrix in the network $W_i(t) = \sum_{j=1}^{R} \sigma_j(t) u_j(t) v_j(t)^\top$ at training time t, we compute

$$S(t)_{jk} = |\langle u_j(t) v_j(t)^\top, u_k(T) v_k(T)^\top \rangle|, \tag{5}$$

where $T$ is the final step of training, and the absolute value is taken to ignore sign flips in the SVD computation. We then plot the diagonal of this matrix $S(t)_{ii} \; \forall \; i \leq 100$ over time. We also use a scalar measure of the diagonal to summarize like in the alignment case: $s(t) = \frac{1}{10} \sum_i S(t)_{ii}$. In Figure 11, we see that top singular vectors converge in direction earlier than bottom vectors.

## B.2 Lottery Tickets Preserve Final Top Singular Vectors

As large singular vectors will become stable late in training, we wonder about the connection to magnitude pruning and the lottery ticket hypothesis. Frankle & Carbin (2018) first showed evidence for the lottery ticket hypothesis, the idea that there exist sparse subnetworks of neural networks that can be trained to a comparable performance as the full network, where the sparse mask is computed from the largest magnitude weights of the network at the end of training. Frankle et al. (2020) build further on this hypothesis and notice that, for larger networks, the masking cannot begin at initialization, but rather at some point early in training. Still, the mask must come from the end of training.

The reason for this particular choice of mask may be connected to the dynamics we previously observed. Specifically, at the end of training large singular values are disproportionately larger, so high-magnitude weights may correspond closely to weights in the top singular vectors. If magnitude masks were computed at the beginning, the directions that would become the top singular vectors might be prematurely masked as they have not yet stabilized, which may prevent learning on the task.

Here we train an unmasked VGG-16 (Simonyan & Zisserman, 2014) on CIFAR10, then compute either a random mask, or a global magnitude mask from the end of training, and rewind to an early point (Frankle et al., 2020) to start sparse retraining. We also do the same with an LSTM (Hochreiter & Schmidhuber, 1997b) on LibriSpeech (Panayotov et al., 2015). Please see Appendix D for details. In Figures 12 and 13, we plot the singular vector agreement (SVA, Eqn. 5) between the final model, masked and unmasked, where we see exactly that magnitude masks preserve the top singular vectors of parameters, and with increasing sparsity fewer directions are preserved. Even though prior work has remarked that it is possible to use low-rank approximations for neural networks (Yu et al., 2017), and others have explicitly optimized for low-rank lottery tickets (Wang et al., 2021; Schotthöfer et al., 2022), we rather are pointing out that the magnitude pruning procedure seems to recover a low-rank approximation though it was intended to be "unstructured."

We also compute the singular vector agreement (SVA) between the masked model trajectory and the original unmasked model trajectory (diagonal of Eqn. 5). We see in Figures 12 and 13 that there is no agreement between the bottom singular vectors at all, but there is still loose agreement in the top singular vectors. Thus, it seems the mask allows the dynamics of only the top singular vectors to remain similar, which we know are most important from the pruning analysis in Figure 4.

Preserving top singular vectors by pruning seems like a natural outcome of large matrices regardless of the mask, so as a control, we follow exactly the same protocol except we generate the mask randomly with the same layerwise sparsity. We can see in Figures 12 and 13 that this results in much lower preservation of top singular vector dynamics, and also performs worse, as in (Frankle et al., 2020). It would not be surprising that random pruning is worse if simply evaluated at the end of training, but masking is applied quite early in training at epoch 4 of 164 long before convergence, so it's striking that the network now fails to learn further even though it is far from convergence. We interpret this as evidence that the mask has somehow cut signal flow between layers, so it is now impossible for the network to learn further, while magnitude pruning and rewinding still allows signals to pass that eventually become important. Examining the SVD dynamics of neural network weights allows us to see all of this structure that was hidden previously.

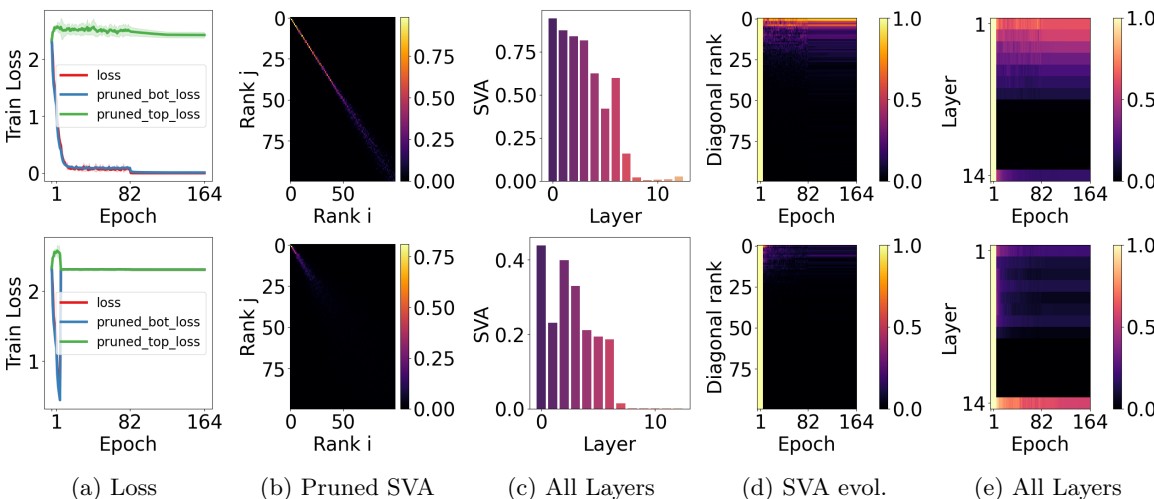

(a) Loss     (b) Pruned SVA     (c) All Layers     (d) SVA evol.     (e) All Layers

Figure 12: Pruning results for VGG. **Top row:** Magnitude pruning. **Bottom row:** random pruning. **First column:** Training loss. We see that at 5% sparsity magnitude pruning is significantly better than random pruning of the same layerwise sparsity. **2nd column:** Singular vector alignment pre- and post-pruning at the end of training for a single layer (the 3rd convolution). We see that magnitude pruning approximates many more top singular vectors, than random pruning at the same sparsity. **3rd column:** Singular vector alignment score pre- and post-pruning across all layers. Agreement is higher across all layers for magnitude pruning, though curiously deeper layers do not agree, likely as later layers are wider so weights are lower magnitude and more will be pruned by the unstructured process. **4th column:** Singular vector alignment between the pruned and unpruned models along the training trajectory. We see that the magnitude pruning still has similar dynamics in its top singular vectors, while random pruning does not. **Last column:** Singular vector alignment score between pruned and unpruned models across layers and time. Again evolution is similar for early layers with magnitude pruning, and completely different for random pruning.

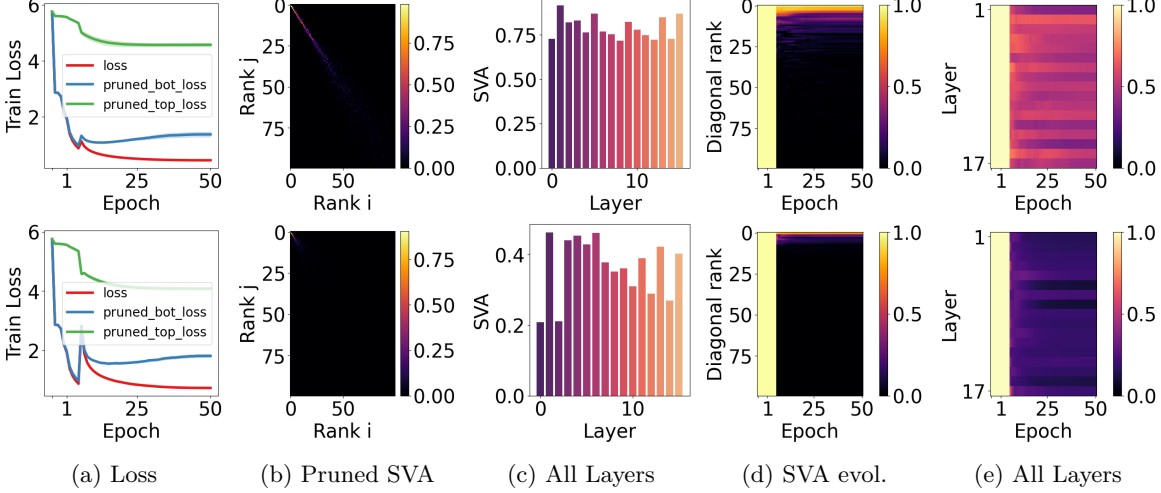

(a) Loss     (b) Pruned SVA     (c) All Layers     (d) SVA evol.     (e) All Layers

Figure 13: Pruning results for LSTM. **Top row:** Magnitude pruning. **Bottom row:** random pruning. See Figure 12 for details. Results are quite similar for the LSTM at 25% pruning as the VGG in Figure 12.

### B.3 Spectral Dynamics and Linear Mode Connectivity

We come to the final phenomenon that we seek to describe: linear mode connectivity. Linear mode connectivity (LMC) is the property that one can interpolate linearly between two different minima in weight space and every parameter set along that path performs well, which gives the impression that the loss surface of

neural networks is somehow convex despite its theoretical nonconvexity. This was first demonstrated in small networks with the same initialization (Nagarajan & Kolter, 2019), then expanded to larger networks and connected to lottery tickets (Frankle et al., 2020; Paul et al., 2022). Entezari et al. (2021) first conjecture that two arbitrary minima show LMC up to permutation, and demonstrate it in simple models. Others expanded this to wide models (Ainsworth et al., 2022; Jordan et al., 2022; Qu & Horvath, 2024), and permutation connectivity can be proven in various ways (Kuditipudi et al., 2019; Brea et al., 2019; Simsek et al., 2021; Ferbach et al., 2023). Still, these conditions and results do not hold for standard models (Qu & Horvath, 2024). LMC has also been exploited for model-averaging and performance gains (Wortsman et al., 2022; Ilharco et al., 2022; Rame et al., 2022). Still despite all of this work, we lack a description for why LMC occurs. In particular: why is there a convex, high dimensional (Yunis et al., 2022) basin that models find shortly in training (Frankle et al., 2020), or after pretraining (Neyshabur et al., 2020; Sadrtdinov et al., 2023)? We do not answer this question in full, but find a deep connection through the dynamics of singular vectors.

### B.3.1 Linear Mode Connectivity Correlates with Top Singular Vector Agreement

We saw in Figure 11 that top singular vectors converge in direction earlier. We also know that for models to display LMC, they need to share an early part of the training trajectory. Perhaps the top singular vectors become stable after this early stage, so we might expect mode-connected solutions to share these components. To examine this, we plot agreement between the singular vectors of the weight matrices at either endpoint of branches:

$$W^{(1)}(T) = \sum_j^R \sigma_j(T) u_j(T) v_j(T)^\top \ ,$$

$$W^{(2)}(T) = \sum_k^R \sigma_k'(T) u_k'(T) v_k'(T)^\top \ ,$$

spawned from the same initialization in training. If the branches are split from an initialization on a trunk trajectory $W(t)$, we call $t$ the split point or epoch. We visualize the diagonal of $|\langle u_j(T) v_j(T)^\top, u_k'(T) v_k'(T)^\top \rangle|_{jk}$ vs. split epoch, where the absolute value is taken to ignore sign flips in SVD computation.

The technical definition of LMC requires measuring the bump, or barrier, in the loss surface along the linear interpolation between final checkpoints(Frankle et al., 2020). To measure this precisely, we use the definition from Neyshabur et al. (2020), which is the maximum deviation from a linear interpolation in the loss, an empirical measure for convexity in this linear direction. When this deviation is 0, we consider the checkpoints to exhibit LMC. Please see Appendix D.10 for details on the calculation. Given evidence in Figure 4 that top components are the most important for prediction, and that top components become stable before training has finished, it is plausible that LMC is connected to the stability of top singular vectors in the later portion of training.

This would mean that checkpoints that do not exhibit the LMC property should not share top singular vectors, while checkpoints that do exhibit the LMC property should share top singular vectors. We see in Figure 8 that this is the case across models and tasks, where the alignment between endpoints is much stronger in top singular vectors. We also see no LMC and poor agreement in top components between branches that have initializations from different trunk trajectories, but with the same split epoch $t$ and the same branch data order in Figure 14. Thus, these top directions are not a unique property of the architecture and data, but rather are dependent on initialization. It is notable that concurrent work (Ito et al., 2024) arrives at a similar conclusion: permutation solvers between optima match top singular vectors. Though the conclusions are similar, their experiments are primarily conducted on smaller scale settings, and only for permutation matching at the end of training. Here we connect these observations to the optimization behavior of SVDs throughout training, tying our previous observations on generalization in with weight averaging.

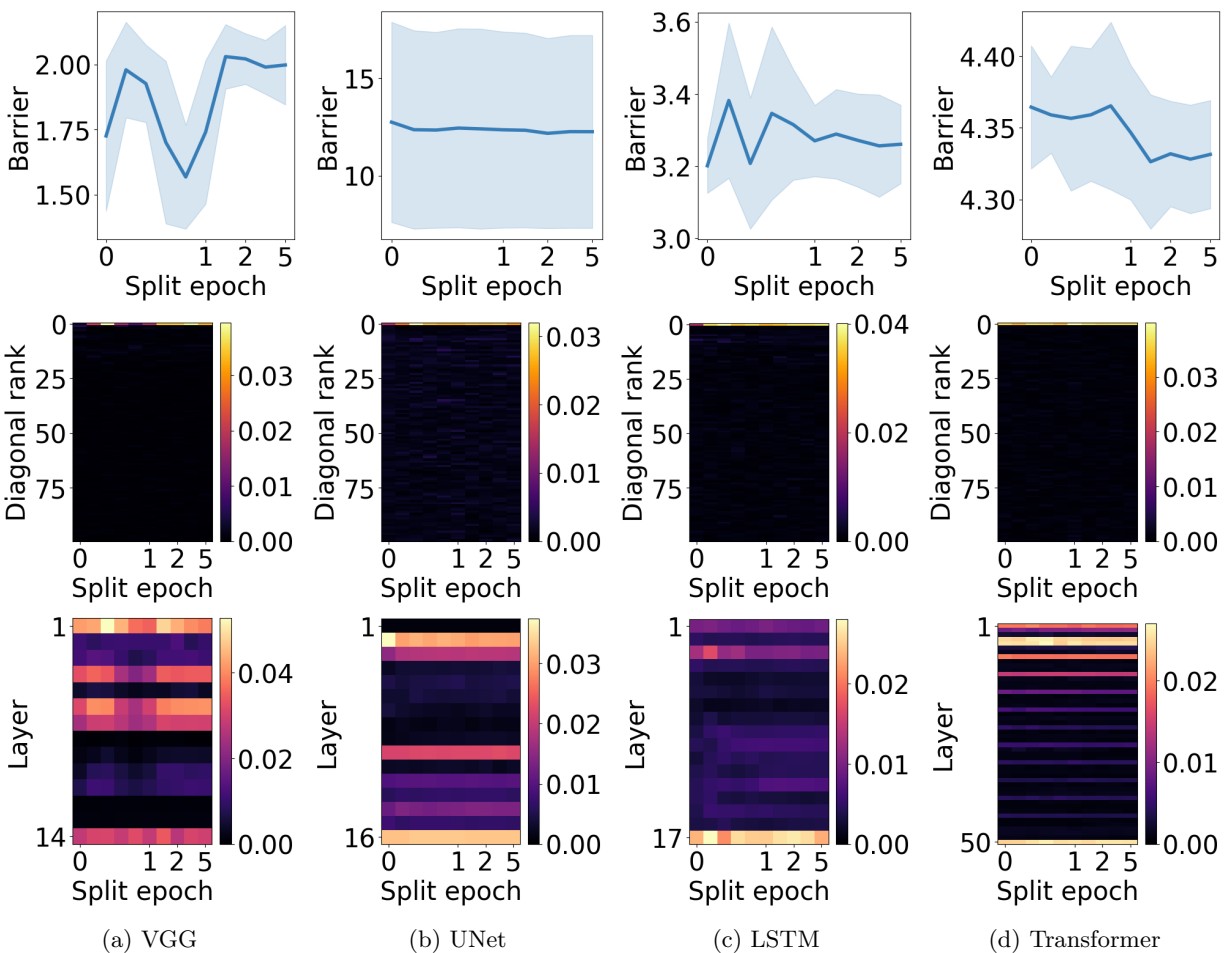

Figure 14: **Top row:** Barrier size vs. split step. **Middle row:** singular vector agreement for a single matrix parameter between branch endpoints that do not share a common trunk, but do share split time and branch data order. **Bottom row:** summary statistic for singular vector agreement across layers. We see that when branches do not share a common trunk, there is neither LMC nor singular vector agreement, even though the optimization is otherwise the same.

### B.3.2 Perturbing Breaks Linear Mode Connectivity and Singular Vector Agreement Simultaneously

To make the connection between top singular vectors and LMC even tighter, we intervene in the training process. If we add random perturbations to destabilize the components that will become the top components long before they have converged, and if singular vector agreement is tied to LMC, we would like to see that final models no longer exhibit the LMC property. Indeed this is the case. In Figure 15, when increasingly large random perturbations are applied, the barrier between final checkpoints increases and the LMC behavior disappears. Please see Appendix D for details. In addition, the previously-strong singular vector agreement disappears simultaneously. Thus it seems this agreement is tied to linear mode connectivity.

We speculate that, due to the results in Figure 4 that show the top half of the SVDs are much more critical for performance, if these components are shared then interpolating will not affect performance much. Rather, interpolation will eliminate the orthogonal bottom components which may only make a minor impact on performance. If however the top components are not shared, then interpolating between two models will remove these components, leading to poor performance in between. Such observations may help in explaining the utility of pretraining (Neyshabur et al., 2020), weight averaging (Rame et al., 2022; Wortsman et al.,

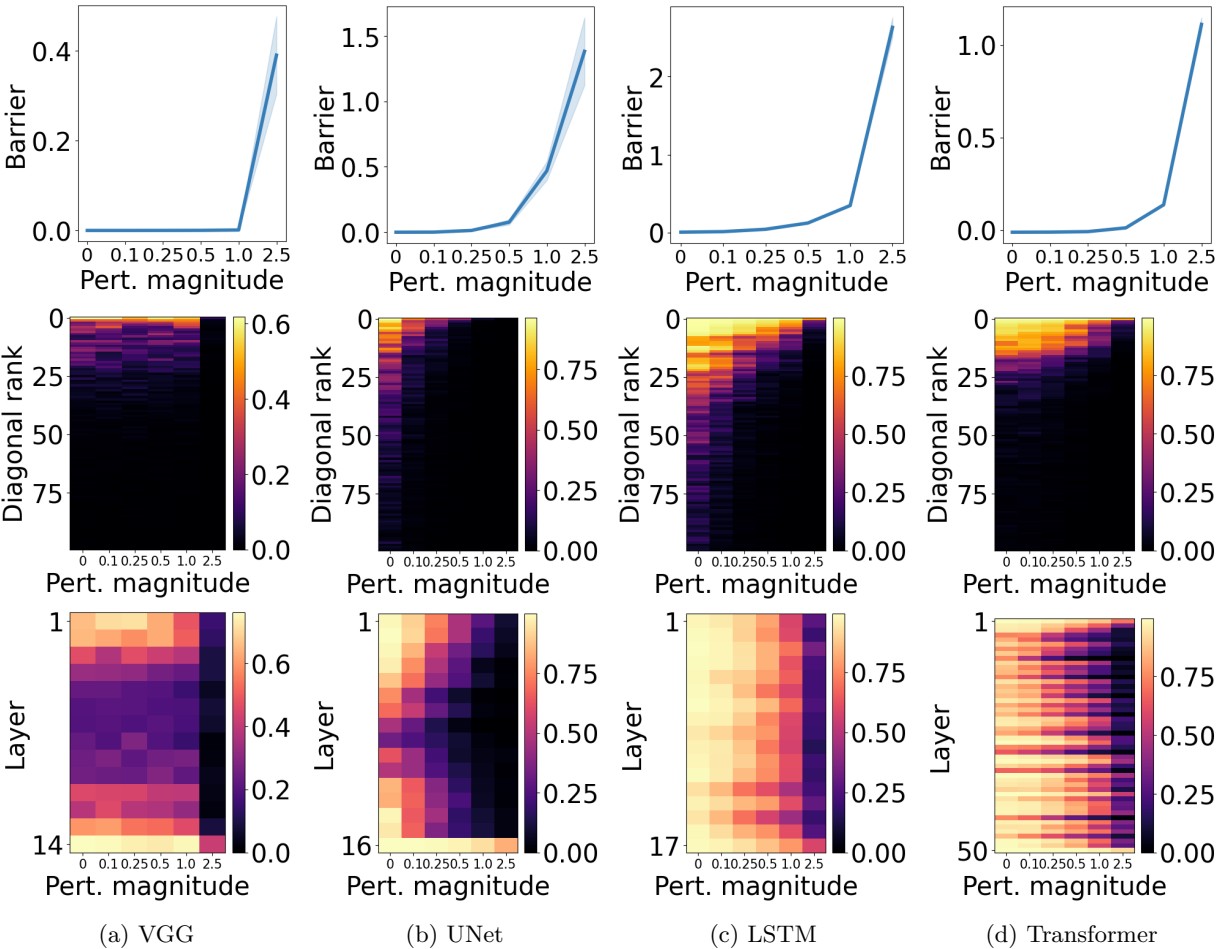

Figure 15: **Top row:** Barrier size vs. perturbation magnitude. **Middle row:** singular vector agreement for a single matrix parameter between branch endpoints vs. perturbation magnitude. **Bottom row:** summary statistic for singular vector agreement across layers with perturbation magnitude. We see that whereas without perturbation models would exhibit LMC after training, with increasing perturbations the LMC property disappears simultaneously with the agreement in top singular vectors.

2022; Ilharco et al., 2022) or the use of LoRA (Huh et al., 2022) to replace full finetuning. Studying spectral dynamics allows us to see this deep structure.

## C   Explanation of Balancedness

Prior work on deep linear networks (Arora et al., 2019; Milanesi et al., 2021) suggests that rank minimization may describe implicit regularization in deep matrix factorization better than simple matrix norms. See Arora et al. (2018) (Appendix A) for a detailed argument. However, a critical assumption used in these works is "balanced initialization." This means that for consecutive matrices $W_i$ and $W_{i+1}$ in the product matrix $\prod_j W_j$, we have $W_{i+1}^\top W_{i+1} = W_i W_i^\top$ at initialization. Decomposing these matrices with SVDs and leveraging orthogonality, this simplifies to $V_{i+1} \Sigma_{i+1}^2 V_{i+1}^\top = U_i \Sigma_i^2 U_i^\top$ where $U_i$ and $V_{i+1}$ are orthogonal matrices. Since these are orthogonal decompositions of the same matrix, their diagonals must be equivalent, allowing for the permutation of elements with the same value. This leads to $U_i = V_{i+1} O$ up to signs, where $O$ is a block diagonal permutation matrix that may permute the rows of equivalent diagonal elements. Notably, if all diagonal elements are distinct and $U_i$ and $V_{i+1}$ are square matrices, then $U_i = V_{i+1}$ up to signs. This gives us matching singular vectors for consecutive matrices.

# D   Experimental Details

For all experiments, we use 3 random seeds and average all plots over those 3. This is relatively small, but error bars tend to be very tight, and due to the high volume of runs required for this work we lack the resources to run much more.

In order to compute alignment we consider only pairs of consecutive layers that directly feed into each other, and ignore the influence of residual connections so as to cut down on the number of comparisons. Specifics on individual architectures are given below.

## D.1   Image Classification with VGG

We train a VGG-16 (Simonyan & Zisserman, 2014) on CIFAR-10 (Krizhevsky, 2009) for 164 epochs, following the hyperparameters and learning rate schedule in (Frankle et al., 2020), but without data augmentation as it contributes extra randomness. For the optimizer we use SGD with batch size 128, initial learning rate 0.1 and momentum of 0.9. We also decay the learning rate 3 times by a factor of 10 at epoch 82, epoch 120, and finally at epoch 160. We also use a minor amount of weight decay with coefficient 0.0001.

VGG-16 uses ReLU activations and batch normalization (Ioffe & Szegedy, 2015), and includes both convolutional and linear layers. For linear layers we simply compute the SVD of the weight matrix. For convolutional layers, the parameters are typically stored as a 4D tensor of shape $(c_{\text{out}}, c_{\text{in}}, h, w)$ for the output channels, input channels, height and width of the filters respectively. As the filters compute a transformation from each position and input channel to an output channel, we compute the SVD of the flattened tensor of shape $(c_{\text{out}}, c_{\text{in}} \cdot h \cdot w)$, which maps all inputs to outputs, similar to Praggastis et al. (2022). This is not the SVD of the entire transformation of the feature map to the next feature map Sedghi et al. (2018), but rather the transformation from a set of adjacent positions to a particular position in the next layer computed by the parameters. For the individual SV evolution plot, we use the 12th convolutional layer.

In order to compute alignment of bases between consecutive convolutional layers, $V_{i+1}^{\top} U_i$ we need to match the dimensionality between $U_i$ and $V_{i+1}$. For convolutional layers we are presented with a question as to how to handle the spatial dimensions $h$ and $w$ as naively the input dimension of the next layer will be a factor of $h \cdot w$ larger dimension. We experimented with multiple cases, including aligning at each spatial position individually or averaging over the alignment at all spatial positions, and eventually settled at aligning the output of one layer to the center spatial input of the next layer. That is, for a 3x3 convolution mapping to a following 3x3 convolution, we compute the alignment only for position (1,1) of the next layer. This seemed reasonable to us as on average the edges of the filters showed poorer alignment overall. For the individual alignment plot, we use the alignment between the 11th and 12th convolutional layers at the center spatial position of the 12th convolutional layer.

## D.2   Image Generation with UNets

We train a UNet (Ronneberger et al., 2015) diffusion model (Sohl-Dickstein et al., 2015; Ho et al., 2020) on MNIST (LeCun, 1998) generation. We take model design and hyperparameters from (Wang & Vastola, 2022). In particular we use a 4-layer residual UNet and train with AdamW (Loshchilov & Hutter, 2017) with batch size 128, and learning rate of 0.0003 for 100 epochs. This model uses swish (Ramachandran et al., 2017) activations and a combination of linear and convolutional, as well as transposed convolutional layers.

Computing SVDs and alignment is similar to the image classification case described above, except in the case of the transposed convolutions where an extra transpose of dimensions is needed as parameters are stored with the shape $(c_{\text{in}}, c_{\text{out}}, h, w)$. For the individual SV evolution plot, we use the 3rd convolutional layer. For the alignment plot, we use the alignment between the 3rd and 4th convolutional layers at the center spatial position of the 4th convolutional layer.

### D.3    Speech Recognition with LSTMs

We train a bidirectional LSTM (Hochreiter & Schmidhuber, 1997a) for automatic speech recognition on LibriSpeech (Panayotov et al., 2015). We tune for a simple and well-performing hyperparameter setting. We use AdamW (Loshchilov & Hutter, 2017) with batch size 32, learning rate 0.0003 and weight decay 0.1 for 50 epochs. We also use a cosine annealing learning rate schedule from 1 to 0 over the entire 50 epochs.

The LSTM only has matrix parameters and biases, so it is straightforward to compute SVDs of the matrices. For individual SV evolution plots, we plot the 3rd layer input parameter. In the case of alignment, we make a number of connections: first down depth for the input parameters, then connecting the previous input parameter to the current hidden parameter in both directions, then connecting the previous hidden parameter to the current input parameter. In particular the LSTM parameters are stored as a stack of 4 matrices in PyTorch, and we find alignment is highest for the "gate" submatrix, so we choose that for all plots. For the individual layer alignment, we plot alignment between the 3rd and 4th layer input parameters.

### D.4    Language Modeling with Transformers

We train a Transformer (Vaswani et al., 2017) language model on Wikitext-103 (Merity et al., 2016). We base hyperparameter choices on the Pythia suite (Biderman et al., 2023), specifically the 160 million parameter configuration with sinusoidal position embeddings, 12 layers, model dimension 768, 12 attention heads per layer, and hidden dimension 768. We use AdamW (Loshchilov & Hutter, 2017) with batch size 256, learning rate 0.0006 and weight decay 0.1. We use a context length of 2048 and clip gradients to a maximum norm of 1. We also use a learning rate schedule with a linear warmup and cosine decay to 10% of the learning rate, like Biderman et al. (2023).

For SVDs, for simplicity we take the SVD of the entire $(3d_{\mathrm{model}}, d_{\mathrm{model}})$ parameter that computes queries, keys and values from the hidden dimension inside the attention layer, without splitting into individual heads. This is reasonable as the splitting is done after the fact internally. We also take the SVD of the output parameters, and linear layers of the MLPs, which are 2 dimensional matrices. For the individual SV evolution plot, we plot the SVs of $W_1$ of the 8th layer MLP

For alignment, we consider the alignment of $W_Q$ and $W_K$ matrices, $W_V$ and $W_O$ matrices, computing alignment between heads individually then averaging over all heads. We also consider the alignment between $W_O$ and $W_1$ of the MLP block, between $W_1$ and $W_2$ of the MLP block, and between $W_2$ and the next attention layer. For the individual layer alignment, we plot alignment between $W_1$ and $W_2$ of the 8th layer MLP.

### D.5    Spectral Dynamics with Scale (Pythia)

Here we apply the perspective developed in Section 5 to larger scale models. As we lack the resources to train these models ourselves, we leverage the Pythia (Biderman et al., 2023) family which provides training trajectories for language models across a range of scales (70m to 12b parameters). We are further constrained to the 2.8b parameter model at the largest due to memory requirements when computing SVDs and alignment.

In Figure 16, we see similar rank dynamics across a variety of scales. We choose to select the 7th layer MLP to compare between models as it is present at all scales. We do see an unequal evolution in singular values, but also a contraction as training proceeds for longer. The difference between scales is not very obvious, but proportionally fewer of the singular values evolve to be large in the 2.8b model as opposed to the 410m model, which one can see from the thickness of the light magenta color. The lack of alignment except for the top rank is quite consistent with earlier observations, and such alignment happens much later for the largest model.

### D.6    Weight Decay Experiments

All tasks are trained in exactly the same fashion as mentioned previously, with increasing weight decay in the set $\{0, 0.0001, 0.001, 0.01, 0.1, 1.0, 10.0\}$. For ease of presentation we consider a subset of settings across

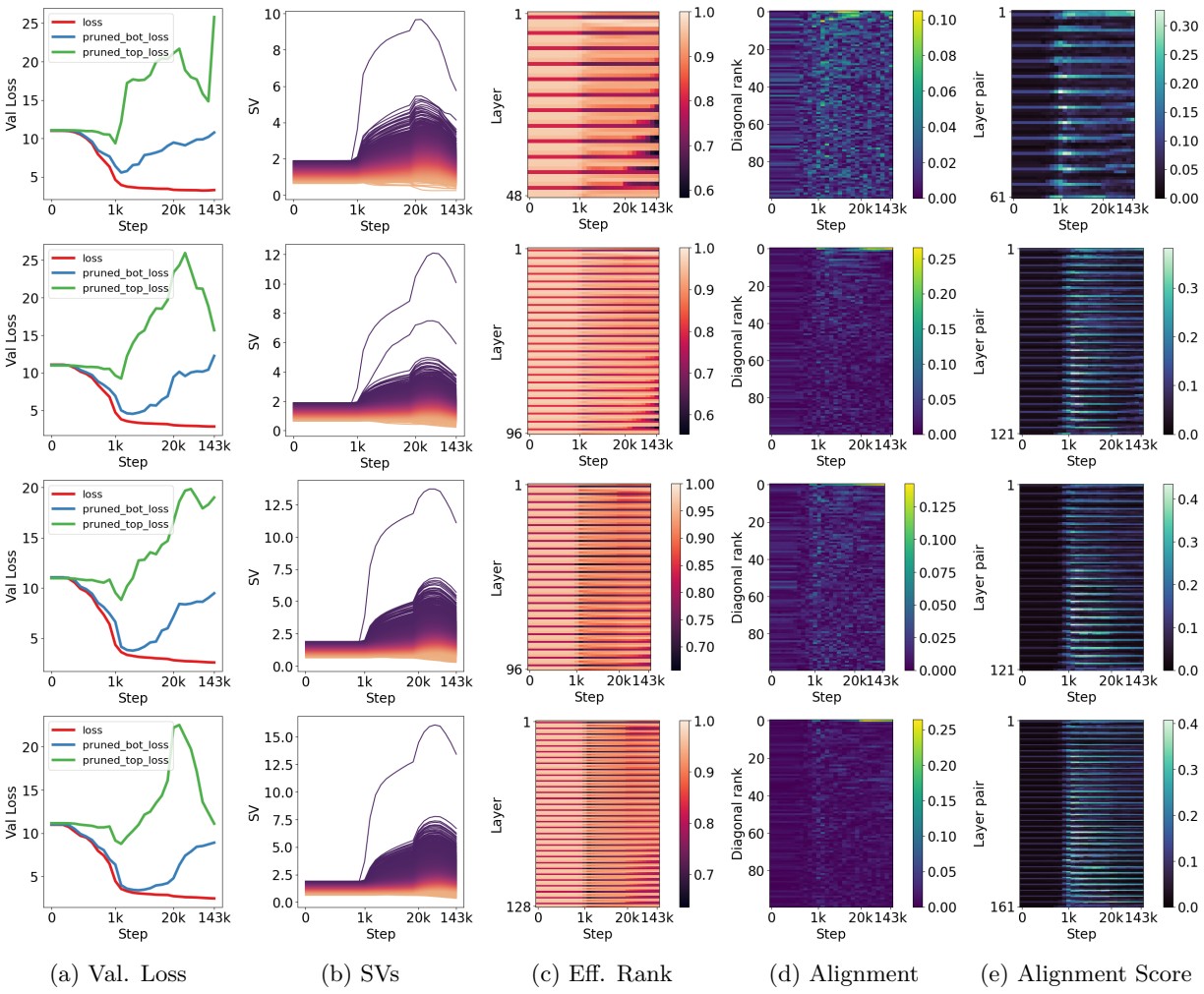

(a) Val. Loss  (b) SVs  (c) Eff. Rank  (d) Alignment  (e) Alignment Score

Figure 16: Spectral dynamics of Pythia suite. From top to bottom we examine the 160m, 410m, 1.4b and 2.8b parameter models. Notably, much less noise appears in the alignment plot with increasing scale. Presumably this could be due to the fact that larger dimensional vectors have higher probability to be orthogonal, which may play a role in making optimization easier. We see stronger alignment score (Eqn. 4) in all layers in the larger model, perhaps because of that cleaner signal.

tasks. In Figure 18 we include trained model performance and pruned model performance to show that, even with high levels of weight decay, models do not entirely break down. More so, the approximation of the pruned model to the full model gets better with higher weight decay.

## D.7 Grokking Experiments

For the Trasnformer, we mostly follow the settings and architecture of Nanda et al. (2023), except we use sinusoidal positional encodings instead of learned.

For the slingshot case we follow hyperparameter settings in Thilak et al. (2022), Appendix B except with the 1-layer architecture from Nanda et al. (2023) instead of the 2-layer architecture specified. We perform addition modulo 97. The original grokking plot in Thilak et al. (2022) appears much more dramatic as it log-scales the x-axis, which we do not do here for clarity.

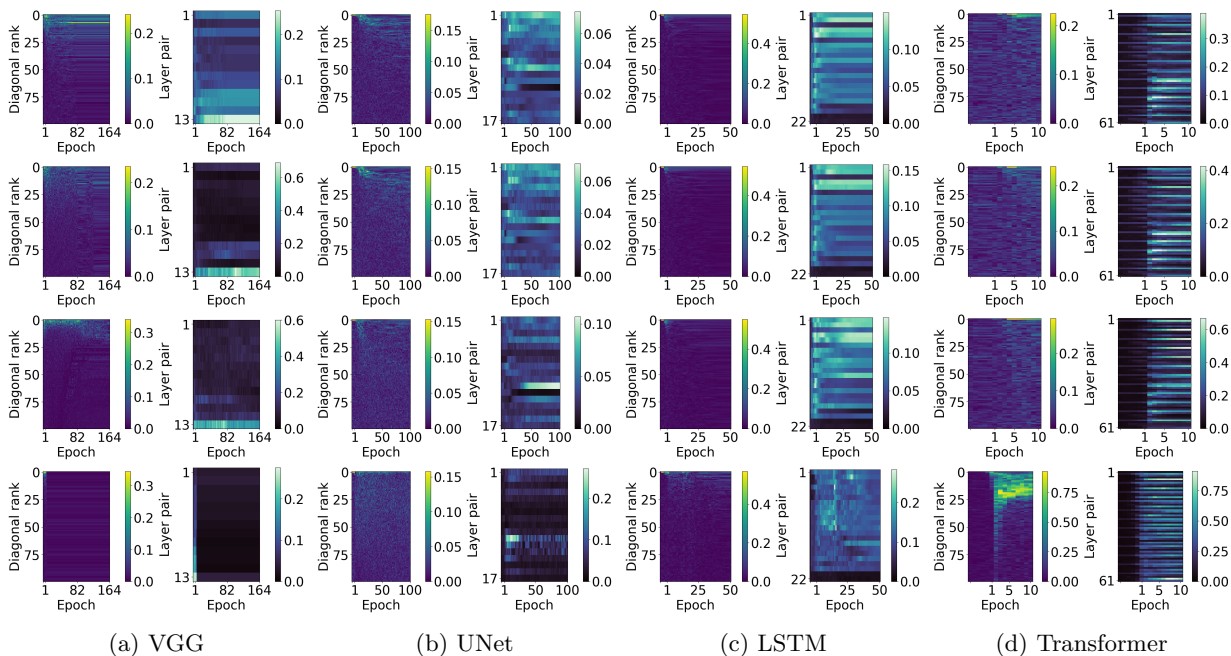

(a) VGG          (b) UNet          (c) LSTM          (d) Transformer

Figure 17: Diagonal of alignment for a single pair over time (Eqn. 3) and alignment metric across pairs of matrices over time (Eqn. 4) where the y-axis represents depth. From top to bottom, for VGG we use coefficients $\{0, 0.001, 0.01, 0.1\}$, while for other networks we use coefficients $\{0, 0.1, 1, 10\}$. We see that the maximum alignment magnitude is higher with large weight decay, and in particular, the Transformer has the strongest alignment even when nonlinearities separate the MLP layers.

In the case of the deep MLP, we follow Fan et al. (2024), where we use a 12-layer MLP with ReLU activations and width 400, trained on MSE loss on MNIST (LeCun, 1998). We use 2000 examples, a batch size of 100, weight decay 0.01, and initialization scale 8 (Liu et al., 2023).

### D.8 Random Label Experiments

We train a 4-layer MLP on CIFAR10 (Krizhevsky, 2009) with either completely random labels, or the true labels. We use SGD with momentum of 0.9 and constant learning rate of 0.001, and train for 300 epochs to see the entire trend of training. The major difference to the setting of Zhang et al. (2021) is the use of a constant learning rate, as their use of a learning rate schedule might conflate the results.

For the VGG case, we follow our previous hyperparameters, except we leave out weight decay and learning rate scheduling, instead using a constant learning rate of 0.01.

For the LSTM case, we follow our previous hyperparameters, and extend the training budget to 200 epochs allow for the random label setting to train longer. In this case, our network does not have sufficient capacity to memorize the data completely.

### D.9 Magnitude Pruning Experiments

We use the same VGG setup as described previously. In this case we train until the end, then compute a global magnitude mask. To do this we flatten all linear and convolutional weights into a single vector, except for the last linear layer, and sort by magnitude. Then we keep the top 5% of weights globally, and reshape back to the layerwise masks. This results in different sparsity levels for different layers, so when generating the random masks, we use the per-layer sparsities that resulted from the global magnitude mask.

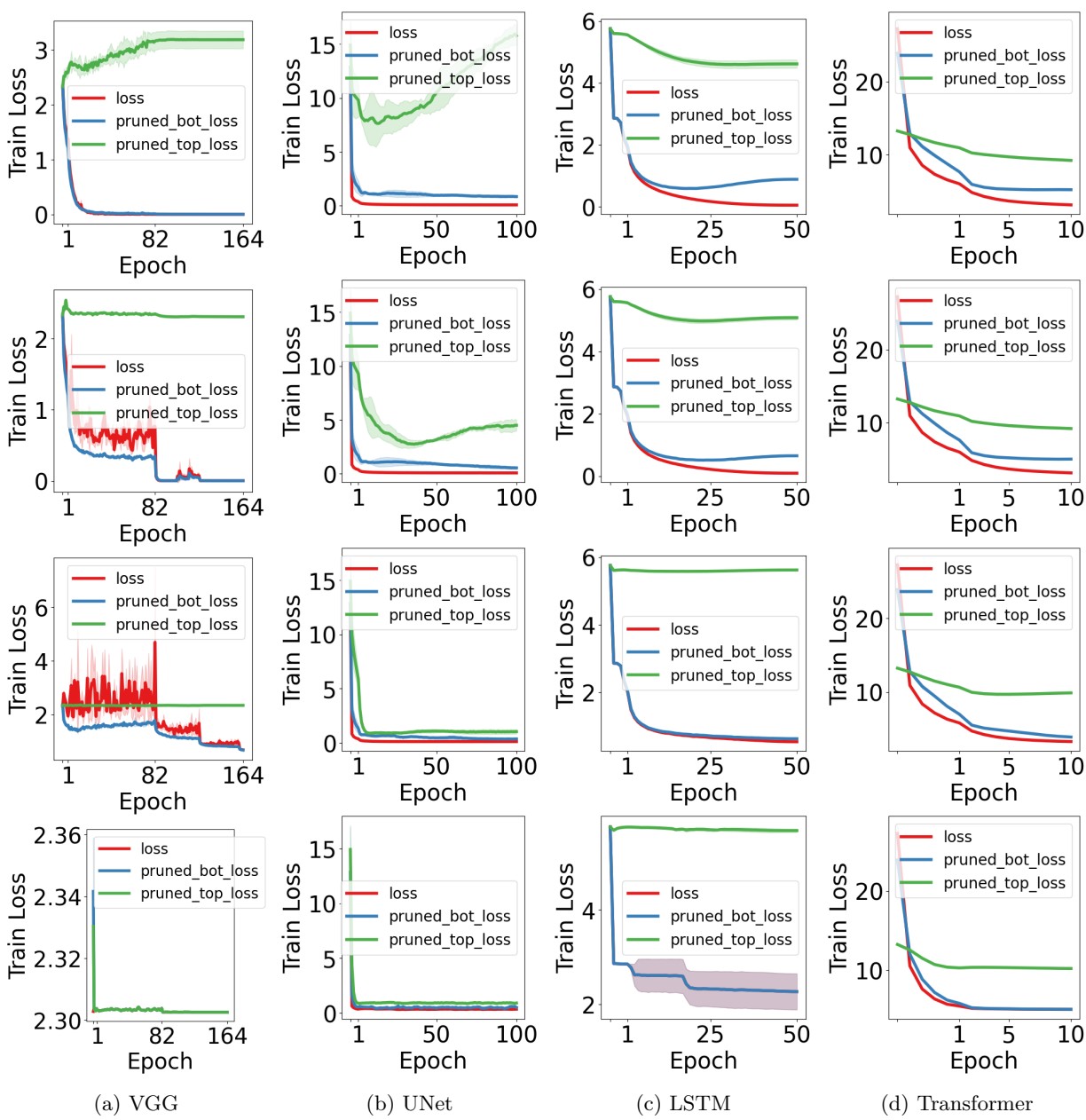

Figure 18: Training loss over time, where the rows use differing amounts of weight decay. From top to bottom, for VGG we use coefficients $\{0, 0.001, 0.01, 0.1\}$, while for other networks we use coefficients $\{0, 0.1, 1, 10\}$. We see that it is still possible to achieve low training loss under high weight decay, and as we increase the amount of weight decay, the gap between pruned and unpruned parameters closes, lending support to the idea that the parameters become lower rank.

To retrain the network, we rewind to epoch 4 (after the point of LMC (Frankle & Carbin, 2018)), then continue training with the mask, always setting other weights and their gradients to 0. We average all results over 3 random seeds.

For the LSTM we follow exactly the same procedure, except our mask only reaches a level of 25% sparsity, due to larger performance degradations with higher sparsity.

### D.10    LMC Experiments

We save 5 evenly-spaced checkpoints in the first epoch, as well as at the end of the next 4 epochs for 10 intializations in total. We train 3 trunks, and split 3 branches from each trunk for a total of 9 branches which we average all plots over.

Following Neyshabur et al. (2020), we compute the barrier between checkpoints as follows: given $W^{(1)}(T)$ and $W^{(2)}(T)$ that were branched from $W(t)$ we compute

$$b(t) = (\max_{\alpha \in [0,1]} \mathcal{L}((1-\alpha)W^{(1)}(T) + \alpha W^{(2)}(T)) - ((1-\alpha)\mathcal{L}(W^{(1)}(T)) + \alpha \mathcal{L}(W^{(2)}(T)))) \tag{6}$$

when this quantity is 0, we consider the checkpoints to exhibit LMC.

We recompute batch normalization parameters after interpolating for VGG-16, and group normalization parameters for the UNet, as these do not necessarily interpolate well (Frankle et al., 2020). We also compute singular vector agreement for the same parameter between either branch endpoint.

To plot the singular vector (dis)agreement and LMC between different modes, we make 11 evenly spaced measurements interpolating between branch endpoints that had the same split epoch, and the same branch seed, but different trunk initializations.

### D.11    Perturbed LMC Experiments

We perturb all weights $W$ after the point of dynamics stability where we expect to see LMC at the end of training (epoch 4 is sufficiently late in all cases) using randomly sampled normal perturbations $\epsilon \sim \mathcal{N}(0, I)$ with $\|\epsilon\| = \eta\|W\|$ where $\eta \in \{0.0, 0.1, 0.25, 0.5, 1.0, 2.5\}$. We do not perturb the output layer, as this has a very substantial effect on the optimization. We also do not perturb the input layer for the Transformer as it is too computationally expensive for our resources.

## E    Limitations

There are a few key limitations to our study. As mentioned, we lack the computational resources to run more than 3 random seeds per experiment, though we do find error bars to be quite tight in general (except for the generalization epoch in the grokking experiments). In addition, as discussed we ignore 1D parameters like biases and normalization in the neural networks, which may play additional roles. Due to computational constraints we do not consider alignment of layers across residual connections as this quickly becomes combinatorial in depth, thus there may be other interesting interactions that we do not observe. Finally, due to computational constraints we are unable to investigate full results on larger models than the 12 layer Transformer, which may have different behavior, but the results on the Pythia suite in Appendix D.5 are encouraging.

## F    Additional Experiments

### F.1    Effect of initialization scale

In Figure 20 we explore the effect of initialization scale. Prior work (Woodworth et al., 2020) showed that in deep linear systems the choice of initialization modulated the effect of incremental learning of ranks, with smaller initialization increasing this effect. We observe that alignment is slightly stronger with smaller initialization, mirroring the theory in deep linear networks.

### F.2    Effect of learning rate

In Figure 21 we explore the effect of learning rate on dynamics for an MLP trained on CIFAR10. Ghosh et al. (2025) proved that balancedness may develop in deep linear systems when the learning rate is sufficiently large. We see here that the trend is mixed: learning rate 0.01 shows the strongest balancedness, and going above or below has a weaker effect.

### F.3 Individual seed plots for grokking

For clarity, in Figure 22 we show an average of 3 seeds of grokking modular addition with Transformers (see Figure 2). The transition to low error and low rank coincide.

### F.4 Full alignment matrix evolution

We previously plotted only the diagonal of the alignment matrix (Eqn. 3) instead of considering off-diagonal elements. In Figure 23 we provide the evolution of the entire alignment matrix between two layers throughout training for an MLP on CFIAR10 so as to explain why: anecdotally we did not observe much signal off-diagonal, hence the focus on the upper diagonal in Eqns 3 and 4.

### F.5 Image classification

In Figure 24 we provide the behavior for all networks tested in CIFAR10 side-by-side to facilitate comparison. We see across architectures that there is qualitative agreement in the trend toward rank minimization.

### F.6

## G Compute Resources

All experiments are performed on an internal cluster with on the order of 100 NVIDIA 2080ti GPUs or newer. All experiments run on a single GPU in less than 8 hours, though it is extremely helpful to parallelize across machines. We estimate that end-to-end it might take a few days on these resources to rerun all of the experiments in this paper. Additionally, the storage requirements for all of the checkpoints will take on the order of 10 terabytes.

## H Code Sources

We use PyTorch (Paszke et al., 2019) and NumPy (Harris et al., 2020) for all experiments and Weights & Biases (Biewald, 2020) for experiment tracking. We make plots with Matplotlib (Hunter, 2007) and Seaborn (Waskom, 2021). We also use HuggingFace Datasets (Lhoest et al., 2021) for Wikitext-103 (Merity et al., 2016).

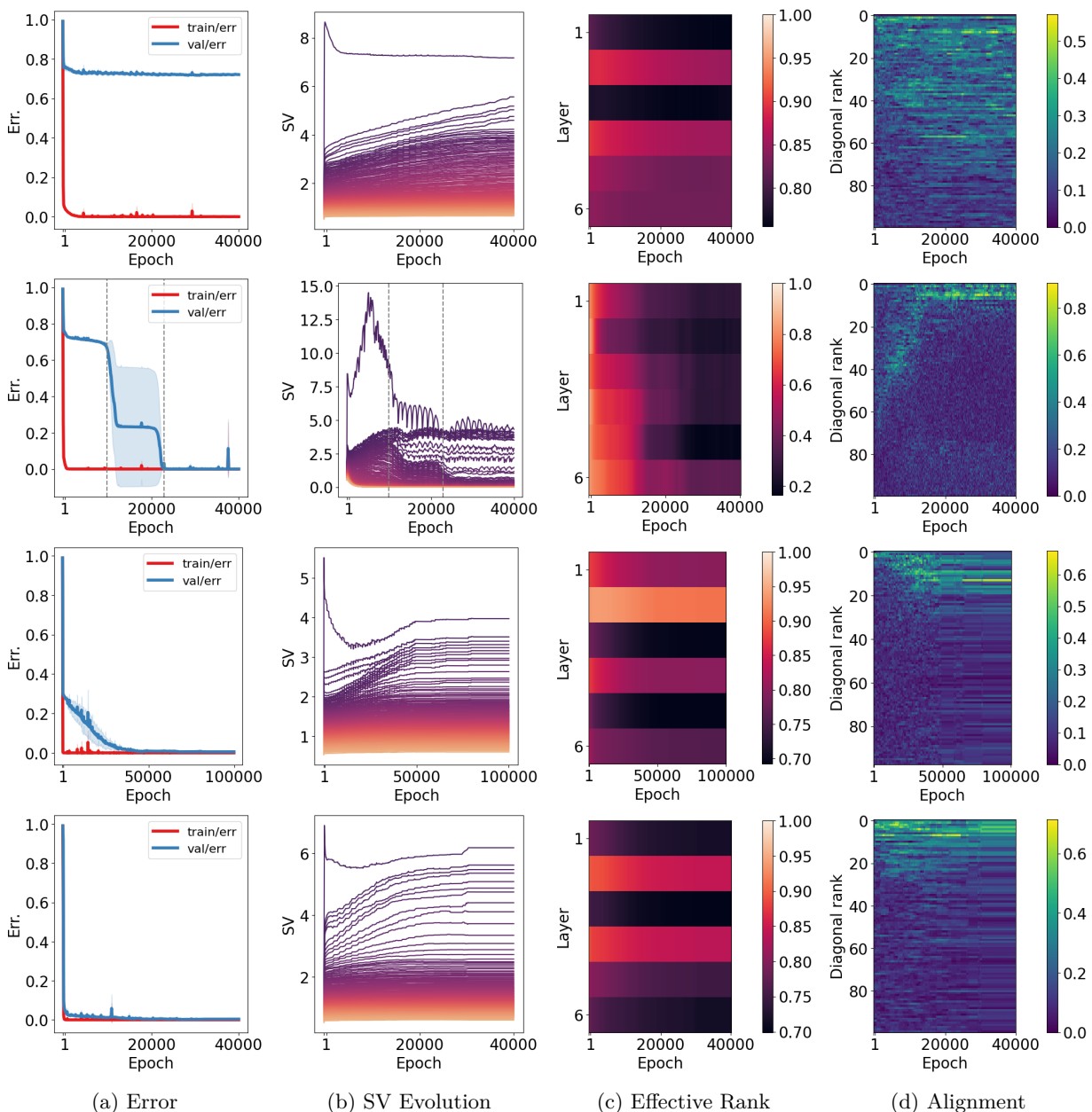

(a) Error      (b) SV Evolution      (c) Effective Rank      (d) Alignment

Figure 19: **Grokking and Spectral Dynamics in Modular addition. Top row:** 30% data and no weight decay. **2nd row:** 30% data and weight decay 1.0 (grokking), using hyperparameters from Nanda et al. (2023). **3rd row:** 70% data with no weight decay (slingshot), using hyperparameters from Thilak et al. (2022). **Bottom row:** 90% data and no weight decay. **1st column:** Training and validation error. **2nd column:** Singular value evolution is visualized for the first attention parameter, where each line represents a single singular value and the color represents the rank. **3rd column:** Effective rank of all layers (Eqn. 1). **4th column:** Alignment (Eqn. 3) between the embedding and the first attention parameter is also visualized, where the y-axis corresponds to index $i$ of the diagonal. One can see that grokking co-occurs with low-rank weights. In addition, there is an alignment that begins early in training that evolves up the diagonal. Without weight decay and with less data, neither grokking nor the other phenomena occur during the entire training budget, but using more data, even without weight decay, leads to low-rank solutions from the beginning of training. The slingshot case follows a similar trend, though the validation loss is gradually fit. Across cases with good generalization, parameters are lower rank, and alignment is also more prevalent in the top ranks.

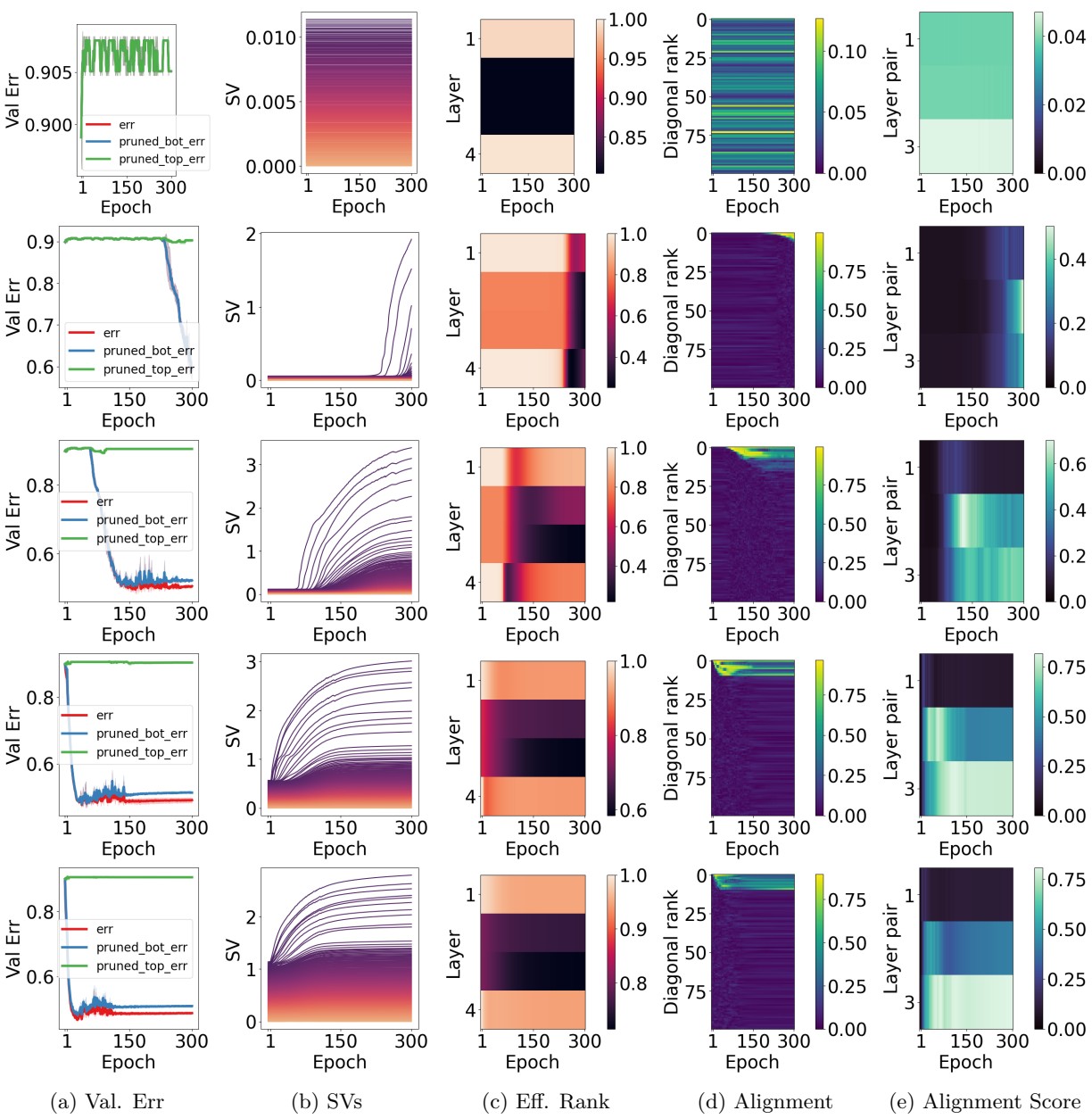

(a) Val. Err  (b) SVs  (c) Eff. Rank  (d) Alignment  (e) Alignment Score

Figure 20: Varying initialization scale affects the dynamics of an MLP trained on CIFAR10. From top to bottom we use default PyTorch initialization multiplied by the constants $\{0.001, 0.01, 0.05, 0.1, 0.5, 1.0\}$. Notably, alignment occurs slightly more strongly with smaller initialization, as predicted by Woodworth et al. (2020) for deep linear models.

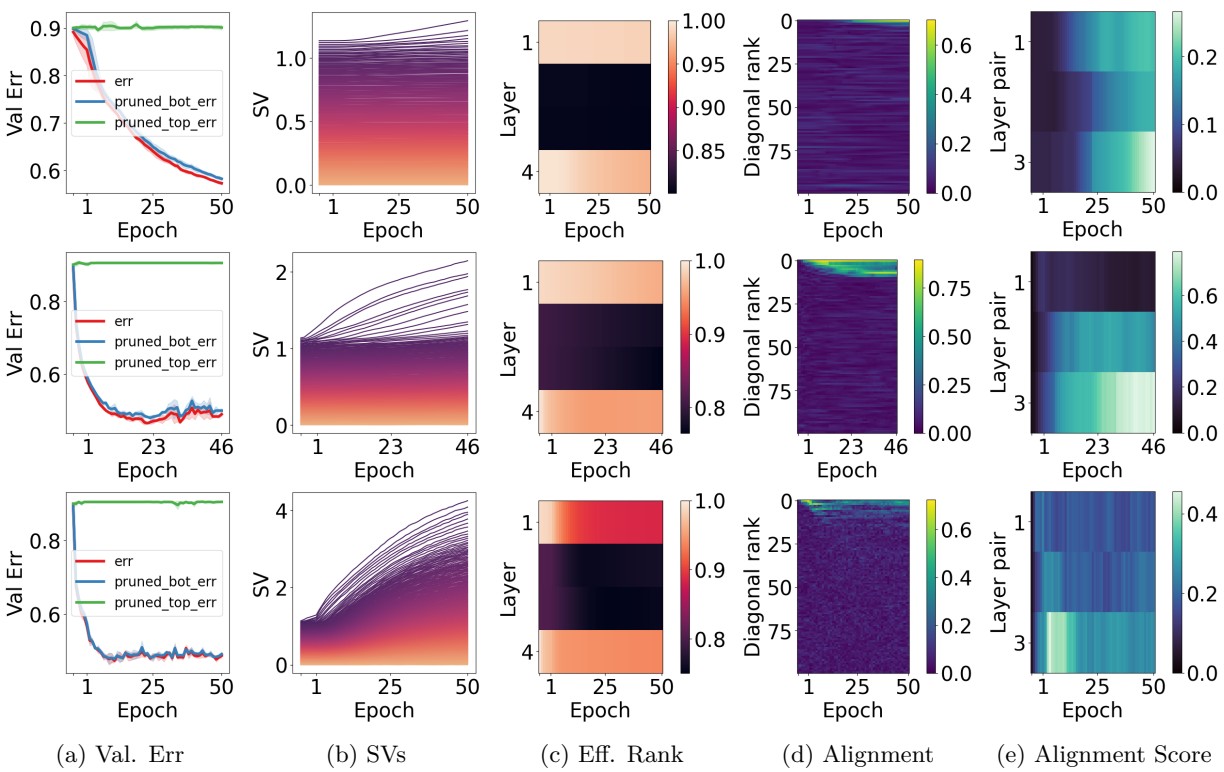

(a) Val. Err    (b) SVs    (c) Eff. Rank    (d) Alignment    (e) Alignment Score

Figure 21: The effect of varying learning rate on an MLP trained on CIFAR10. From top to bottom we use learning rates $\{0.001, 0.01, 0.1\}$. Notably, alignment increases with larger learning rate, agreeing with the deep linear setting in Ghosh et al. (2025).

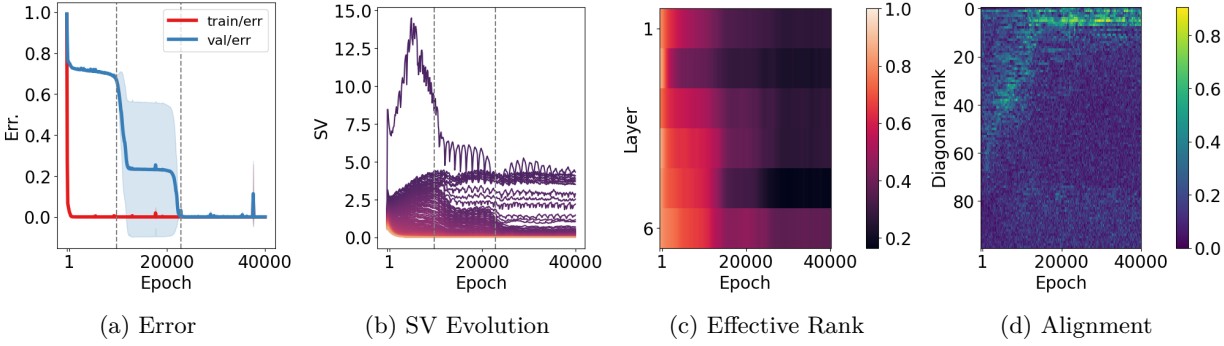

(a) Error    (b) SV Evolution    (c) Effective Rank    (d) Alignment

Figure 22: Plots for averages of 3 seeds of grokking modular addition with Transformers (Figure 2). We see a stark transition to low-rank when validation error decreases.

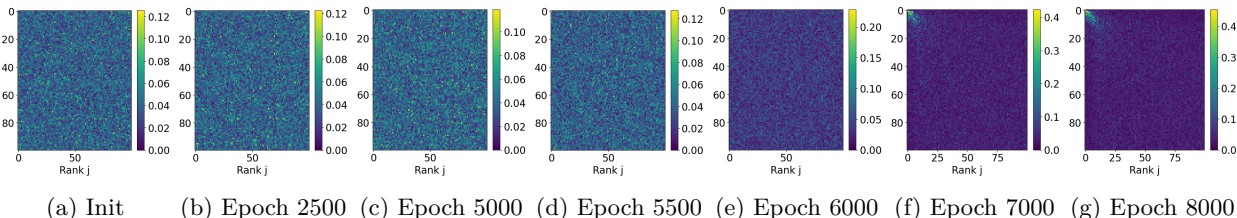

(a) Init    (b) Epoch 2500    (c) Epoch 5000    (d) Epoch 5500    (e) Epoch 6000    (f) Epoch 7000    (g) Epoch 8000

Figure 23: Alignment matrix (Eqn. 3) over time. We see that the majority of signal is concentrated in the diagonal, an example of why we consider the diagonal in the rest of the work.

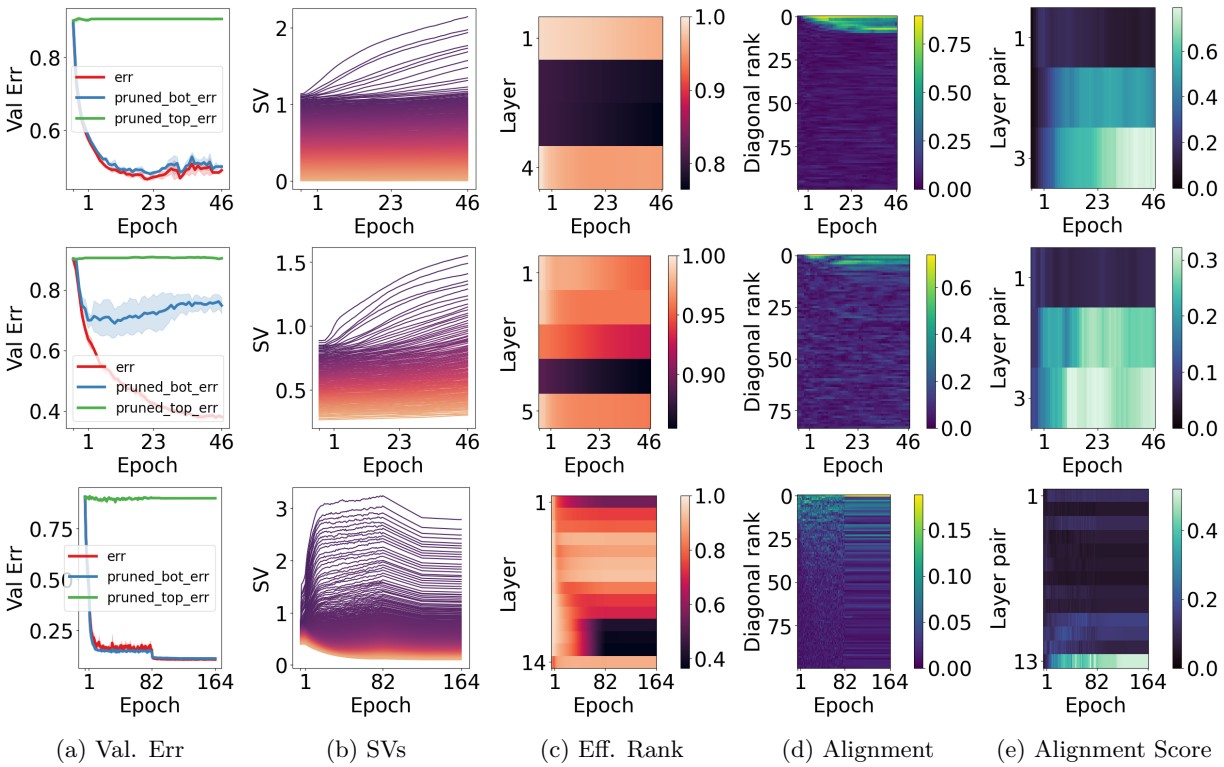

(a) Val. Err     (b) SVs     (c) Eff. Rank     (d) Alignment     (e) Alignment Score

Figure 24: Different architectures all trained on CIFAR10. From top to bottom, MLP, LeNet-5 and VGG-16. We see roughly consistent spectral dynamics across architectures within the same task.

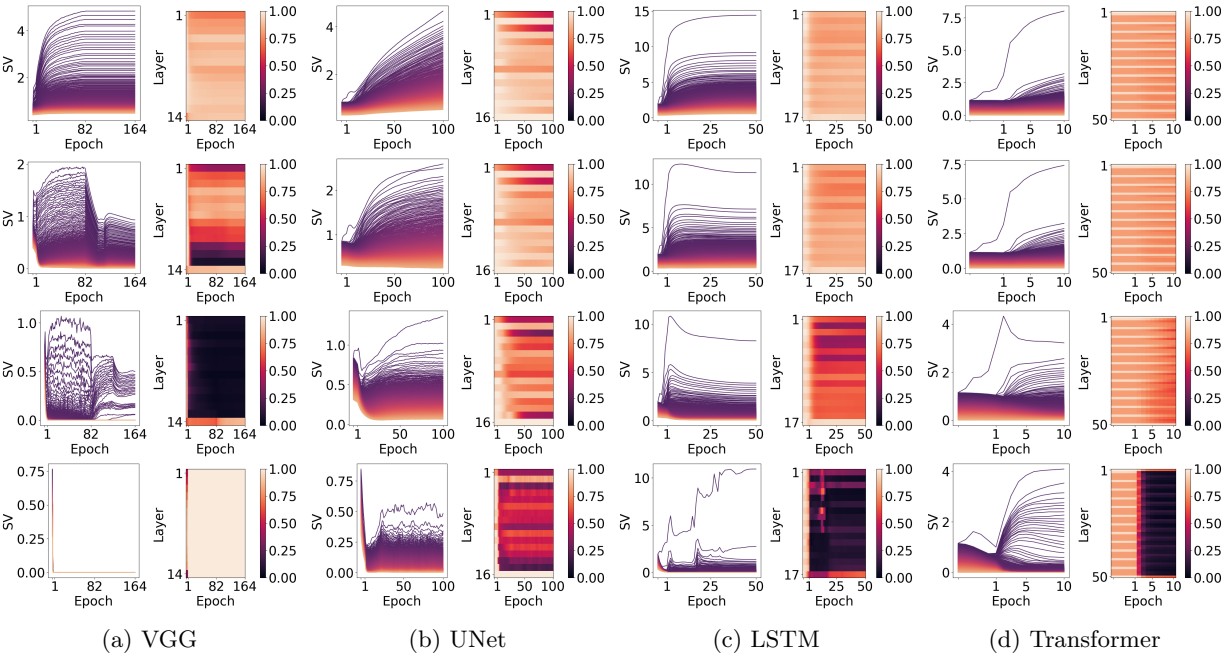

(a) VGG     (b) UNet     (c) LSTM     (d) Transformer

Figure 25: Duplicate of Figure 6 with shared color scaling between settings.

