# OpenReview forum: "Approaching Deep Learning through the Spectral Dynamics of Weights"
_TMLR — Rejected by TMLR_

### Review · Reviewer_pZQf · 2025-04-16

**Summary Of Contributions:**

This submission presents an empirical study on the spectral dynamics of neural network weights. That is, during training, the singular value decomposition of the weight matrices are computed. Specifically, the value of all singular values are tracked, as well as alignment scores between singular vectors of consecutive layers. It is found that certain pattern in the spectral dynamics correlate with phenomena that so far lack a comprehensive theoretical explanation, e.g., crocking, lottery tickets or generalization despite memorization. Moreover, it is found that there is a bias towards rank minimization during training.

**Audience:**

Yes

**Broader Impact Concerns:**

There are no concerns on the ethical implications of this work that require adding a Broader Impact Statement.

**Claims And Evidence:**

No

**Requested Changes:**

## Critical

- To address the first weakness, the experimental evaluation needs to cover more datasets, architectures, and training setups. As the authors already mentioned their limited computational ressources, it might be neccessary to narrow the scope of the submisison, e.g. to "image classification with VGGs", "image classification with convolutional networks, "speech recognition with LSTMs", etc. While this would certainly limit the submission in terms of the evaluation criterion "Would at least some individuals in TMLR's audience be interested in knowing the findings of this paper?", meeting the "Are the claims made in the submission supported by accurate, convincing and clear evidence?" is of higher importance.

- The paper’s structure needs improvement for clarity. Currently it is difficult to find information as often times it located in an unexpected place. For example, the effective rank and alignment scores, i.e., quantities that are central to all experiments are defined in section 3 (Grokking and Rank Minimization). I also recommend to directly state at the bottom of page 3 the location of the there stated. As already noted, I was unable to find where it is shown that "Top singular vectors are preserved when performing magnitude pruning".

- The presentation of the empirical results needs to be adapted so that conclusions can be made more easily.
Currently, it is difficult to infer whether rank minimization is happening from the barplots in column (c, effective rank). Using a more contrastive colormap might be a solution if one wants to stick with the barplots. This point also includes the problematic averaging in the grokking experiments, see weaknesses.

- The relevance of alignment for non-linear networks needs to be established and a more informative alignment score needs to be used, see weaknesses.

## Non-critical
- Currently, the singular values of the filters are studied (i.e. matrices of shape $c_{out} \times c_{in} · h · w$). To make the experiments more comaprable across convolutional and linear layers (across models and within the VGG architecture), it would be more appropriate to study the singular dynamics of the linear map between entire images that performed by the convolutional layers. The latter can be efficiently computed without instantiating the corresponding matrix, see [1].
- I would be curious about the authors perspective on related work in [2]. Their experiment (see table 1) show that in image classification limiting the spectral norms of layers during training improves the generalization capabilities of neural networks (validation accuracy is preserved but testing acuracy decays).
Notably, therein spectral norms of layers are controlled by computing the svd (using [1]) and clamping the spectral values to a predefined interval $[0,t]$. Without accounting for second order effects, this approach should increase the entropy of singular value distributions and therefore *increase the effective rank*.

[1] Sedgie et al., The singular values of convolutional layers, ICLR 2018
[2] Graf et al., On Measuring Excess Capacity in Neural Networks, NeurIPS 2022

**Strengths And Weaknesses:**

# Strengths
- The submission studies the singular dynamics of neural networks across a wide range of tasks (image classification, image generation, speech recognition and language modelling) and phenomena (memorization vs. generalization, lottery tickets, linear more connectivity).
- Moreover, it identifies the pattern of rank minimization (of weight matrices) during neural network training in the spectral dynamics. This pattern appears to be consistent across models and tasks and can therefore serve as both (i) an assumption for theoreticians studying properties of trained neural networks, (ii) empirical (counter-) evidence for theoretical models of neural network training.

# Weaknesses
- While the submission covers a wide range of tasks and phenomena, empirical depth is limited, as each task uses only a single architecture, dataset, hyperparameter setting, with results averaged over 3 seeds. **This design prevents determining whether the phenomena are general patterns or whether they are dataset/model-specific.**
While random seeds likely have minor impact, I would expect architecture and dataset difficulty and choice of optimizer (parameters) significantly affect singular dynamics.
- Below, I will comment on the claim (from the bottom of page 3) that lack sufficient support.
  - *"Grokking is intimately linked to rank minimization"*
  Specifically, it is claimed that *"the sudden drop in validation loss coincides precisely with the onset of low-rank behavior in the singular values"*
  If this was true, then it should be possible to extract the epoch where the validation loss drops just from the columns (b) sv Evolution, (c) effective rank and (d) alignment.
  I am not able to do so, and I don't think that it can be done. Even if there was enough signal in the data, the circumstance that all plots are averages but the epoch where the loss drop occurs varies drastically between the random seeds makes it impossible.
  - *"Weight decay, norm regularization, enhances the rank minimization behavior"*
  The results presented in Figure 6 are mixed. In some cases, i.e., VGG (0 vs 0.001), LSTM and Transformer (both 0.1 vs 1), stronger weight decay results in higher effective ranks.
  Moreover, the range of only four tested weight decay values is too small to sufficiently support the claim.
  - *"Top singular vectors are preserved when performing magnitude pruning and while linearly interpolating between connected modes."*
  I cannot comment on the linear mode connectivity experiments as I was unable to clearly link the connection between the experimental setup described in D.10 and the results presented in Figure 8. I also did not find a comparison of top singular values before and after pruning. Both might be on my part though.
- The alignment experiments are unconvincing.
  1. The alignment score is motivated by theoretical work on **linear networks** where alignment has a clear meaning as it ensures that features (determined by the svd of the first layer) are propagated independently through the network. In non-linear networks activation functions will break this property and therefore the interpretability of alignment.
  Consequently, the following statement is unjustified: *"Although this behavior echoes theoretical predictions in the deep linear setting, we find that the behavior of networks disagrees with a common theoretical assumption about low-rank dynamics: the alignment of singular vectors in consecutive layers (Saxe et al., 2014; Arora et al., 2018; 2019; Milanesi et al., 2021). Thus, the rank minimization mechanism differs from what the theory describes."*
  This is not a contradiction between theory and practice, but simply a consequence of comparing the theory of linear networks with experiments on networks with activation functions.
  2. The evaluation score, i.e., the sum of the absolute scalar products of top-10 singular vectors of consecutive layers, ignores that vectors can be aligned but singular values could be order differently. A more informative score would account for such orderings. For example one could take the minimum of the current score over all permutations of singular vectors (i.e., rows in the matrix $A_{jk}$ in Eq. (3).  Potentially, this could explain that *"In the image classification case, we see a similar rank transition, though alignment appears seemingly out of nowhere."*

---

> ### Author Response · Authors · 2025-05-01
> **Thank you for taking the time to review (1/3)**
>
> Thank you for taking the time to review and for your thorough read. We apologize for the delay in responding, but were unsure whether to wait for a 3rd review before doing so.
>
> Thank you as well for acknowledging the strengths of our work: that the submission covers and links many phenomena, and that the trends we identify are general across tasks. We will respond point by point to your concerns below:
>
> ## Weaknesses
>
> > "empirical depth is limited... design prevents from determining whether the phenomena are general patterns or whether they are dataset/model specific..."
>
> We agree that it is important to be rigorous. For this reason we chose to experiment across both models and tasks. Above in the strengths the reviewer stated that the patterns appear to be consistent across models and tasks. That similar dynamics occur regardless of the architecture/dataset means that the support for these trends are much stronger than an in-depth study on e.g. image classification. This is because the identified bias is not just a property of a particular dataset, or the overparametrization regime with 0 training error or low-rank targets like image classification, but also appears in tasks like language modeling where it is impossible to achieve 0 training error and the targets are quite high rank. We also experiment across optimizers already: VGG uses SGD, LSTM uses Adam, Transformer uses AdamW. The aim of our work is to point out that standard hyperparameter settings drawn from the literature have this bias. It is not our object to induce low-rank bias. The proposed experimental focus on a narrower distribution would *by-definition* be much more dataset specific and thus contradicts the reviewer's own criticism, so it is unclear to us why it sheds any more light than the existing experiments. In any case, if this is unsatisfying we also provide additional experiments on MLP, LeNet and VGG in Appendix F on image classification.
>
> We also want to point out that the existing empirical exploration we present is far above the standard of evidence for similar work. For some examples:
> - Progress Measures [1]: Single-layer Transformers on modular arithmetic
> - Deep Grokking [2]: MLPs on image classification
> - LMC [3]: ConvNets on image classification
> - CKA [4]: ConvNets on image classification
> - Simplicity Bias [5]: MLPs on regression, ConvNets on image classification
> - Why do we need weight decay [6]: ConvNets on image classification, Transformers on language modeling
> - Implicit bias [7]: Deep linear networks on matrix factorization
> - Ours: Transformers on modular arithmetic, MLPs on image classification, ConvNets on image classification, UNet on image generation, LSTM on speech recognition, Transformers on language modeling (even large scale in Appendix D.5)
>
> [1] https://arxiv.org/abs/2405.19454
> [2] https://arxiv.org/abs/2301.05217
> [3] https://proceedings.mlr.press/v119/frankle20a
> [4] https://arxiv.org/abs/1905.00414
> [5] https://arxiv.org/abs/2103.10427
> [6] https://arxiv.org/abs/2310.04415
> [7] https://proceedings.neurips.cc/paper/2019/hash/c0c783b5fc0d7d808f1d14a6e9c8280d-Abstract.html
>
> > "Grokking is intimately linked to rank minimization... it should be possible to extract the epoch where validation loss drops just from the columns... I don't think it can be done"
>
> We are a bit puzzled. One can see that validation error drops for the transformer in two places: epoch 10k and around epoch 23k. In the corresponding figures in column b) and c) we see transitions in the number of active ranks. A similar trend occurs for the MLP at epoch 6k, though it is more visible in column b) due to the relatively large number of singular values compared to the transformer obscuring the trend in rank. To allay concerns of seed averaging, we provide the same plots for a single seed of the Transformer in Appendix F. Does this answer your concern?
>
> > "... The results presented in Figure 6 are mixed"
>
> We appreciate the need for rigor, but we are again a bit confused. Perhaps there is a miscommunication due to the fact that the color bars have different limits. In particular one can see as we descend the rows in Figure 6 that the lower limit of these color bars decreases monotonically, thus the colors are not comparable between plots, and indeed effective rank is decreasing with more weight decay, including for all the cases you provided. Does this answer your concern?
>
> > "I cannot comment on the linear mode connectivity experiments..."
>
> The goal is to understand whether the fact that top singular vector stability (due to unequal SV evolution) is at all connected to model averaging. As such, if we have stable dynamics for a small subset of singular vectors, they should be shared between endpoints displaying LMC. Hence, we visualize the agreement between singular vectors of different endpoints in Figure 8, where we see that low barriers (implying LMC) coincide with singular vector agreement. Does this help?

---

> > ### Comment · Reviewer_pZQf · 2025-05-09
> > **Response to authors**
> >
> > I thank the authors for their detailed responses to my concerns.
> >
> > Below I will comment on their points that did not convince me yet.
> >
> > > This is because the identified bias is not just a property of a particular dataset, or the overparametrization regime with 0 training error or low-rank targets like image classification, but also appears in tasks like language modeling where it is impossible to achieve 0 training error and the targets are quite high rank.
> >
> > My point was that you did not provide enough evidence to establish whether there is such bias in each individual setting, which is a prerequisite for claiming there is a bias across settings.
> >
> > > We also want to point out that the existing empirical exploration we present is far above the standard of evidence for similar work. For some examples:
> >
> > I consider this as an invalid argument, since the referenced works were not published at TMLR whose main acceptance criterion is "Are the claims made in the submission supported by accurate, convincing and clear evidence?" but at venues that permit reviewers to weigh insufficient empirical evidence against strong theoretical contributions and novelty.
> >
> > > To allay concerns of seed averaging, we provide the same plots for a single seed of the Transformer in Appendix F. Does this answer your concern?
> >
> > The provided single seed plot is much clearer and should be moved to the main part of the manuscript together with such a plot for the MLP (it would be even more clearer if you add vertical lines at the epoch you consider to 'coincide precisely with the onset of low-rank behavior'). However, my concerns are not fully answered yet, since you only provided single seed plots for one of the two architectures.
> >
> > > We appreciate the need for rigor, but we are again a bit confused. Perhaps there is a miscommunication due to the fact that the color bars have different limits.
> >
> > I did miss the different limits and I apologize. However, I still cannot confidently assess whether the effective rank decreases with weight decay strength, as I'm unable to mentally recalibrate the color bars. While I am more positive that there is such a trend, only a clearer presentation (e.g. using consistent color bar limits [0,1] across weight decay strengths / all experiments) can fully convince me.
> >
> > > The goal is to understand whether the fact that top singular vector stability (due to unequal SV evolution) is at all connected to model averaging... Does this help?
> >
> > Not really, as I am still puzzled about the experimental setting.

---

> ### Author Response · Authors · 2025-05-01
> **Thank you for taking the time to review (2/3)**
>
> We continue our response to the initial review:
>
> > "I also did not find a comparison of top singular values before and after pruning..."
>
> We believe you are referring to contribution bullet 5 describing top singular vector preservation during pruning. This content is in Section 6 under "Lottery Tickets" but the terminology is not standardized. To aid organization we now link to the appropriate section in the contribution bullets.
>
> > "...In non-linear networks activation functions will break this property and therefore the interpretability of the alignment..."
>
> We agree that alignment is motivated by linear networks and state exactly as much in Sections 2 and 3. There is a long line of methods in deep learning motivated by linear networks, including closely-related orthogonal initialization (http://proceedings.mlr.press/v80/xiao18a). Though on the face of it it may seem that this linear quantity makes no sense for nonlinear networks, we see for certain settings (Transformers/MLPs in Section 3, MLPs in Figure 7, Transformers in Figure 17 column d, bottom row) that there is actually linear alignment *despite* the nonlinearities. Thus it is not so simple as saying that linear alignment is an uninterpretable quantity for nonlinear networks. Certainly such alignment could lead to better compression-based generalization bounds, and help to refine assumptions for theory. That larger scale systems do not agree is a worthwhile contribution for theorists, and something that for example reviewer V5RM finds interesting. If the issue is with the particular phrasing, perhaps we could revise to "...theory describes for linear settings"?
>
> > "The evaluation score... ignores that vectors can be aligned but singular values could be ordered differently..."
>
> We agree that this is a possibility. In our anecdotal experience this did not occur much, hence the choice to work only with diagonals, however this was never explained in the text. To allay concerns we provide a set of plots in Appendix F showing how the off-diagonals tend to look and we've revised the text to reference these plots. There was a pragmatic concern in focusing on the diagonals once we made this observation, as computing such permutations can be quite expensive for each checkpoint of each seed of each run of each experiment.
>
> ## Requested Changes
>
> ### Critical
>
> > "...the experimental evaluation needs to cover more datasets, architectures, and trainings setups..."
>
> We responded above to this criticism, but we believe it is quite unfounded when compared to the standard of evidence of prior work. We do include some additional results on image classification alone in Appendix F. Our goal is not to exhaustively study all hyperparameter settings possible, but rather to sample the literature and show that, across tasks and architectures, there is a common trend in the dynamics of the matrices even though these are very disparate settings.
>
> > "The paper's structure needs improvement for clarity..."
>
> To help these concerns we've linked all mentions of quantities to the equations where they are initially described, and contributions to their respective sections. We also added justification for scores as mentioned above. If there is a particular place where more is needed please let us know. We chose to start with grokking so as to lead the reader through the results in a natural progression, hence the early definition of central quantities.
>
> > "Currently, it is difficult to infer whether rank minimization is happening from the barplots in column (c, effective rank)"
>
> We are unsure what exactly is unclear about the plots. In column c) of Figure 2 the effective rank transitions from near 1 to 0.2, where its limits are from 1 to 0. That is a large change. For the MLP the change is subtler but certainly visible at the first layer, and in column b) for the middle layers (the departure of a small group of lines from the bulk). We have chosen a perceptually uniform colormap with the precise idea that it would be as contrastive as possible. On the point of averaging, please see the single-seed plots in Appendix F.
>
> > "The relevance of alignment for non-linear networks needs to be established..."
>
> See our discussion above and our response to reviewer V5RM. There are certainly readers already interested in how this alignment applies.  On the alignment score point, we hope the additional plots in Appendix F will allay concerns.

---

> > ### Comment · Reviewer_pZQf · 2025-05-09
> > **Respose to authors**
> >
> > >To aid organization we now link to the appropriate section in the contribution bullets.
> >
> > Thank you!
> >
> > > Though on the face of it it may seem that this linear quantity makes no sense for nonlinear networks, we see for certain settings (Transformers/MLPs in Section 3, MLPs in Figure 7, Transformers in Figure 17 column d, bottom row) that there is actually linear alignment despite the nonlinearities.
> >
> > I am puzzled that this is your response to me writing "the following statement is unjustified: "Although this behavior echoes theoretical predictions in the deep linear setting, we find that the behavior of networks disagrees with a common theoretical assumption about low-rank dynamics: the alignment of singular vectors in consecutive layers (Saxe et al., 2014; Arora et al., 2018; 2019; Milanesi et al., 2021).", as the former states there is alignment while the latter states that there is no alignment.
> >
> > > If the issue is with the particular phrasing, perhaps we could revise to "...theory describes for linear settings"?
> >
> > Certainly the phrasing needs to be changed. I am not happy about your suggestion though, as it still indicates that there is some contradiction to the theory while there is none.
> >
> > > To allay concerns we provide a set of plots in Appendix F showing how the off-diagonals tend to look and we've revised the text to reference these plots.
> >
> > Thank you for the plots. They allay my concerns, but do not fully resolve them yet, since the experimental setting (MLP on CIFAR10) differes from the experiment where you observed "a similar rank transition, though alignment appears seemingly out of nowhere." which was an MLP on MNIST.
> >
> > > We responded above to this criticism, but we believe it is quite unfounded when compared to the standard of evidence of prior work.
> >
> > See the first two points above.
> >
> > > We chose to start with grokking so as to lead the reader through the results in a natural progression, hence the early definition of central quantities.
> >
> > My point was not that starting with grokking is bad, but that introducing central quantities in the grokking section makes it difficult to find this information. Please move them to a preliminary section.
> >
> > > We are unsure what exactly is unclear about the plots.
> >
> > This is likely because my comment was unclear and I want to apologize. I intended to refer to all barplots of this kind not only to Figure 2. I agree, that in figure 2 it is clear that rank minimization is happening (while so far it is unclear when exactly, see above). Other Figures are not as clear, e.g. figures 3 and 4 or the left and right column of figure 6.
> >
> > > We have chosen a perceptually uniform colormap with the precise idea that it would be as contrastive as possible.
> >
> > Did you consider a colormap that is the concatenation of a monotone increasing function from 0 to 1
> > with the existing colormap?.
> >
> > > See our discussion above and our response to reviewer V5RM.
> >
> > Unfortunately, I cannot see you response to other reviewers.
> >
> > > The choice we made corresponds to the way gradients flow through the network.
> >
> > Why does your choice correspond to the way the gradient flows through the network, to me it seems that the opposite is the case, since the gradient is still the gradient of the global linear map that is parametrized by the weight tensor.

---

> > > ### Author Response · Authors · 2025-05-27
> > > **Thank you for the continued dialogue (2/2)**
> > >
> > > > "the following statement is unjustified: "Although this behavior echoes theoretical predictions in the deep linear setting..."
> > >
> > > We'll try to recap our argument for the sake of clarity. In our experiments on larger scale systems (Section 4) that the review refers to, there appears to be poor alignment for nonlinear networks. In past theoretical work on low-rank dynamics, assuming alignment/proving alignment is a common prerequisite. The "disagreement" we wrote about refers to the idea that new theory is needed because the assumptions needed for prior theory do not hold trivially in the larger-scale nonlinear setting, even though the conclusions (rank minimization) do.
> > >
> > > We understood the original criticism as follows: it's unclear why we should study linear alignment at all as networks are nonlinear, so there's no reason to expect alignment, and thus no "contradiction" (we never used this word). We responded by pointing out that techniques based in linear networks (e.g. orthogonal initialization) have been useful for deep learning, so we believe it is a good first step to verify whether the linear alignment even extends. We also saw in small-scale cases (Figure 3, 7) and heavy weight decay (Figure 17) that there is linear alignment despite nonlinearities. Thus it is not so simple as to say that nonlinearities = alignment uninterpretable. The distinction between these experiments and those in Section 4 is the choice of hyperparameters and scale of the networks. The specific details of each of these settings are specified in the text already.
> > >
> > > If we misunderstood the initial review please let us know. We hope this clarifies our point.
> > >
> > > > ...it still indicates that there is some contradiction to the theory while there is none
> > >
> > > As we stated above, our goal is to point out that theoretical techniques base themselves in assumptions that cannot be used to describe the empirical behavior of larger scale systems *even though* the outcomes are similar (rank minimization). The disagreement we refer to is that the alignment used for earlier theoretical arguments cannot be relied upon here. Do you have a suggestion as to how to rephrase this? We would be happy to revise it.
> > >
> > > > differs from the experiment where you observed...
> > >
> > > We've changed the setting for the alignment matrix evolution in Appendix F. Hopefully this resolves the concern.
> > >
> > > > I intended to reference all barplots
> > >
> > > We'd like to justify the choices we made a bit. There is a tradeoff to make. We chose to have individual color bars with many plots so as to present the most information in the tight space we have. We believe some of the issues for example in Figure 6 is that there are many plots, so the trends for layerwise effective rank can be quite difficult to view without zooming. We discuss the specifics of resolving this for Figure 4 and 6 below. In general the change in effective rank should be visible due to darkening color per-layer in Figures 4 and 6. That Figure 5 is relatively dark and with unclear patterns is exactly the point, which is already described in the text.
> > >
> > > > "The relevance of alignment for non-linear networks needs to be established" -> "See our discussion above and our response to reviewer V5RM"
> > >
> > > Apologies, it seems the visibility of other reviews happens only during the discussion period after the 3rd review is submitted. We were referencing the fact that the aspect of the paper that V5RM found most interesting was the lack of alignment. Thus the relevance of layerwise alignment is already well motivated for certain communities.
> > >
> > > > "Did you consider a colormap...
> > >
> > > All color maps that we use are perceptually uniform and monotonically increasing 0 to 1, could you specify a little more what you mean? We believe the core issue you're pointing to is that sometimes the color resolution is too compressed per-layer and sometimes it is too large. One way to resolve this would be to normalize the per-layer effective ranks, but that would require a separate color map per layer, which would be impossible to display in the space we have. The additional drawback there would be that it would obscure the absolute differences between different layers (for example the difference between MLPs and Attention layers in Transformers visible in the banding, or between different hidden layer sizes in the VGG). We've added an additional plot for uniform limits across depth in Appendix F.
> > >
> > > > Why does your choice correspond to the way the gradient flows...
> > >
> > > Both objects are very closely related, so there is no substantial disagreement here. Perhaps our terminology was imprecise, but what we mean to say is that ultimately the parameter space is the artifact of the neural network that encodes its behavior, and we are interested in how it evolves. The choice we've made for computing alignment corresponds to the way signal flows between layers with respect to the parameter space. As discussed previously it also lets us compute singular vectors in a tractable way.

---

> ### Author Response · Authors · 2025-05-01
> **Thank you for taking the time to review (3/3)**
>
> We finish our response to the initial review below:
>
> ### Non-critical
>
> > "it would be more appropriate to study singular dynamics of the linear map between entire images"
>
> Thank you for the suggestion! We are aware of this paper and considered it. We believe this is a choice. The choice we made corresponds to the way gradients flow through the network, and we are deeply interested in singular vectors which would be prohibitively expensive to compute for the image transformation.
>
> > "authors perspective on... limiting the spectral norms of layers... improves the generalization"
>
> A very interesting point! Clamping the spectral norm does seem like it would increase rank, but "without accounting for second order effects" is doing some heavy lifting. We're quite curious ourselves about Muon (https://kellerjordan.github.io/posts/muon/) which explicitly has a high-rank update and claims faster generalization. It could be the case that clamping spectral norm leads to decaying singular values due to an effect where small singular values now decay to 0 (see https://proceedings.mlr.press/v201/timor23a.html). In any case this is an interesting target for future work!

---

> ### Author Response · Authors · 2025-05-27
> **Thank you for the continued dialogue (1/2)**
>
> Thanks for the detailed followup. We'll continue the discussion above:
>
> > My point was that you did not provide enough evidence to establish whether there is such a bias in each individual setting, which is a prerequisite for claiming there is a bias across settings
>
> In the strengths, the initial review stated "...it identifies the pattern of rank minimization (of weight matrices) during neural network training in the spectral dynamics. This pattern appears to be consistent across models and tasks" which would appear to us as the reviewer understanding that there is some consistent bias among the settings studied. If this is the case, then it appears to us that the criticism is that we are overclaiming something. Would you be willing to identify the specific section of the text where you believe the overclaiming occurs? Our contribution is stated "Rank minimization is a general phenomenon in more complex tasks and architectures" for which we select the 4 chosen, and the "[consistency]" in the review should mirror the phrasing "general phenomenon in more complex tasks and architectures." Nowhere do we claim that rank minimization always occurs, or for every single setting (indeed there are pathological hyperparameters that would do otherwise). We would be happy to revise the text to match the evidence!
>
> We should also mention, however, that other reviewers specifically wrote: "The experiments are thorough... Due to the variety of the experiments, the conclusions by the authors seem to hold generally," and "The authors perform experiments across a variety of tasks and architectures that make the conclusions more broadly applicable." so we do not believe we are alone in thinking there is support.
>
> > I consider this as an invalid argument, since the referenced works were not published at TMLR...
>
> This is a fair point. However, one of the above works cited was indeed published in TMLR (Low-rank simplicity bias [5], https://arxiv.org/abs/2103.10427). This work makes claims like "we observe that deep nets... are biased to map data into low rank embeddings", "we observe that this bias towards low-rank embeddings exists in a wide variety of common optimizers," "this set of observations leads us to conjecture that deeper networks are implicitly biased to find lower effective rank embeddings because the volume of functions that map to low effective rank embeddings increases with depth." However all of the experiments are in low-rank target settings like image classification or low-rank regression, which might skew the results and conclusions. We have reviewed for TMLR before and believe this paper's experiments were more than enough to make an interesting contribution despite that.
>
> > The provided single seed plot is much clearer...
>
> We've moved the single seed plot up to Figure 2 and moved the averaged plot to the appendix. Regarding the MLP, we didn't think it was necessary to provide single seed plots because the standard deviations are very small in the existing plot (Figure 3, bottom row, shaded regions), indicating that the behavior of each seed is quite similar. Adding additional plots for each seed would not give any more information. If you still think otherwise in light of this, we'd be happy to make another plot.
>
> > I did miss the different limits and I apologize...
>
> We've added a duplicate of the plot with shared color bars in the Appendix and reference it in the text (Figure 25). As you can see it's harder to distinguish individual layer trends with a consistent color bar, which is why we chose to use individual color bars initially.
>
> > Not really, as I am still puzzled about the experimental setting
>
> Would you be willing to clarify where exactly it is confusing? We would be happy to revise the text to clarify.
>
> > My point was not that starting with grokking is bad, but that introducing central quantities in the grokking section makes it difficult to find this information. Please move them to a preliminary section.
>
> Done. Thanks for the suggestion.

---

> ### Author Response · Authors · 2025-06-12
> **Thanks again for your review**
>
> Thank you again for taking the time to review and engage in discussion. We greatly appreciate it. As recommendations will be due, we were hoping to discuss what precise changes/additional experiments you were looking for after our latest rebuttal. Please let us know if there are additional concerns you have after reading it. Thanks again for your time.

---

### Review · Reviewer_V5RM · 2025-04-18

**Summary Of Contributions:**

This work takes an approach to understand several deep learning tasks through the lens of spectral dynamics of weights, i.e, empirical study of evolution of singular values and vectors of weight matrices. The authors state that grokking phenomenon aligns with transition to low rank weights. Furthermore, the authors show that singular values grow unequally and the neighboring layers do not align exactly as theory predicts them to.

Finally the authors show the consequence of inherent rank minimization in tasks such as pruning.

**Audience:**

Yes

**Claims And Evidence:**

Yes

**Requested Changes:**

I suggest the authors to have more convincing experiments on the effect of learning rate and initialization in the practical setup.Since the authors mostly build up on past works, the novelty is very limited. So, proper controlled experiments can further strenghtn the paper.

**Strengths And Weaknesses:**

Strenghts:

1) The experiments are thorough. The paper is written nicely.
2) Due to the variety of the experiments, the conclusions made by the authors seem to hold generally.

Weakness:

1) The novely of the work is very less. Almost all the conclusions made in the paper were previously made by most of the past works. The only thing that stands out is that the authors question the claim of alignment of singular vectors which do not seem to hold exactly in practice. I would suggest the authors to list the novel contributions that were not listed in any past works, so the reader has easy time understanding what it new.

2) *singular values grow unequally*: This phenomenon has been extensively studied in the literature as something called incremental feature learning in Gradient descent. The observations are not something new. Infact the initilization scale plays a big role in the evolution of the singular values. Small initilaization (rich scale) promotes incremental learning whereas large initialization may hamper it. I did not find much emphasis on initialization scale made when this conclusion is made. See for example the theory part in https://arxiv.org/pdf/2002.09277 and then Section-7 of the same paper.

3) *We see a small
amount of alignment in the top ranks between layers shortly after training begins, but this becomes more
diffuse over time.* Does the transition happen when the learning rate scheduler is applied? The figures show a sharp transition when the scheduler is applied. If so, I would suggest the authors to keep the learning rate fixed in the experiments, as alignment of singular vectors heavily depend on the scale of the learning rate.

4) *Assumption on balancedness of singular vectors*: The whole section 2.1 and later in 4.3, the authors state that balancedness assumption is done in all the past literature. As this is a required condition to show alignment and track the singular value dynamics. Although there are works in the literature (https://openreview.net/forum?id=J4Dvxv7WnG) that do not assume balancedness at initialization. However, in such cases, a large learning rate can make the layers balanced. Most importantly, learning rate is plays an important role in both balancedness and alignment of the layers. The lack of experiments using various learning rate in the paper may be a weakness. I would suggest the authors to show some experiments on the effect of learning rate (better without the rate decay) and initialization.

---

> ### Author Response · Authors · 2025-05-01
> **Thank you for taking the time to review**
>
> Thank you for taking the time to review and apologies for our delay. We were unsure whether to wait for a 3rd review before responding jointly but have decided to go ahead.
>
> We appreciate that you highlight our thorough experiments, and that the claims seem to hold well due to the thoroughness of these experiments. We will respond point by point below to concerns:
>
> ## Weaknesses
>
> > 1. novelty of this work is very less
>
> First we should remind the reviewer that perceived novelty is not a criterion for acceptance or rejection in TMLR (https://jmlr.org/tmlr/acceptance-criteria.html) but we believe this to be an unfair characterization. We discuss in detail previous contributions on rank minimization in the introduction and in related work Sections 2.1 and 2.2. If there is a particular place the reviewer feels the discussion is insufficient we would be very happy to revise. In any case, the primary contribution of our work is extending theoretical work in small-scale systems to empirical work on larger neural networks. Thus the scale of Sections 3-5 are novel, and all of the experiments in Section 6 are as well. The listed contributions in the introduction, specifically the phrasing "Rank minimization is a general phenomenon in more complex tasks and architectures" was written to allude to past work.
>
> > 2. ...this phenomenon has been extensively studied in the literature as something called incremental feature learning...
>
> We are aware of prior work on unequal singular value evolution. We cite many prior works in our related work already. We do not intend to present this as never having been studied, but we do believe it is understudied in larger systems that are theoretically intractable. We discuss there and in the introduction ("Although this behavior echoes theoretical predictions in the deep linear setting..."). We even cite quite related work on Greedy Low Rank Learning, but again like the paper you provided these results tend to be for very simple systems. Whether or not they apply to standard hyperparameter settings on tasks more complicated than matrix completion for architectures beyond deep linear networks is not clear. As a result, we do not emphasize initialization scale in the initial paper because we are not interested in inducing low-rank learning via initialization Rather we are interested in the networks and settings that practitioners actually use. However we appreciate the comment, and provide a new experiment in Appendix F on initialization scale for MLPs that shows some similarity to the theory.
>
> > 3. "Does this transition happen when the learning rate scheduler is applied?"
>
> No. One can see the alignment only occurs for the very top rank near the first epoch (small yellow blip in rank 1, Figure 5). Notice also that the color bar scales mean the alignment is quite weak. Learning rate schedule transitions don't happen until much later (epoch 82), and for other networks there is no schedule (MLP, UNet) or the schedule is very gentle and thus negligible early in training (LSTM, Transformer). We appreciate the need for rigor, and would be happy to revise the text of Section 4.3 to make this clearer if you have specific pointers.
>
> > 4. "The whole section 2.1 and later in 4.3, the authors state that balancedness assumption is done in all the past literature"
>
> First, this is incorrect. In Section 2.1 we state "Ji & Telgarsky show that alignment between layers will happen specifically..." we are already aware that not all prior work requires balancedness, but this particular paper required infinite training and it does not look like our results will converge to balancedness any time soon.
>
> > 4. "Although there are works in the literature..."
>
> Thank you for the additional reference! We provide some initial experiments varying the learning rate in Appendix F corresponding to this paper, but the particular paper you provide again concerns itself with deep linear networks, with only some visualizations for sharpness on nonlinear systems. As such we believe we are still in dialogue with the prior literature, and our contributions should be of interest.
>
> > 4. "The lack of experiments using various learning rate in the paper..."
>
> We would again like to point out that our goal is not to induce balancedness (to what end?) but to question how close practical empirical settings are to idealized models in theory. As such it was important for us to select settings that are used by practitioners. Still we hope the additional experiments in Appendix F will answer these concerns.
>
> ## Requested Changes
>
> We provide additional results on MLPs for learning rates and initialization scales in Appendix F. We've already described above why we believe the characterization that the work is not novel is uncharitable, and we welcome further discussion.

---

> ### Author Response · Authors · 2025-06-12
> **Thanks again for your review**
>
> Thank you again for your review. As recommendations will be due soon, please let us know if there are additional concerns after our rebuttal. We hope that we've made it clear the distinction to prior theoretical settings. Thanks again for your time.

---

### Review · Reviewer_7RDY · 2025-05-19

**Summary Of Contributions:**

This paper empirically investigates the dynamics of the eigenspectrum of weights in deep networks across varying tasks and architectures. The authors study the connections between the effective rank and other phenomena in deep networks such as grokking, memorization/generalization, lottery ticket hypothesis, and the effect of weight decay. It is hypothesized that weight decay leads to grokking/improved generalization through reducing rank. The authors also show the lack of a general significant alignment between consecutive layers of a deep network during training, which contradicts certain assumptions in earlier theoretical works.

**Audience:**

Yes

**Broader Impact Concerns:**

Not applicable.

**Claims And Evidence:**

Yes

**Requested Changes:**

It is not clear from Figure 2 that the drop in validation error occurs at the same time as the decrease in the effective rank. In particular, in the second row it seems that the drop in validation error starts before Epoch 5000, while the decrease in the effective rank of the first layer occurs after Epoch 5000. Perhaps the authors can change the visualization to make this more clear, such as using a shared time axis.

**Strengths And Weaknesses:**

The authors perform experiments across a variety of tasks and architectures that make the conclusions more broadly applicable. However, as a consequence of the high number of phenomena that the paper attempts to study, it feels like some connections are not properly established. In particular, the most novel connection in the paper would be between grokking and decrease in rank. However, this is hard to interpret from Figure 2 (see below).

The remaining observations made by the paper are relatively well-known in the literature from both theoretical and empirical points of view, e.g. the connection between weight decay, low effective rank, and generalization. That said, it is still interesting and valuable to show that they consistently hold across different tasks.

---

> ### Author Response · Authors · 2025-05-27
> **Thank you for your review**
>
> Thank you for taking the time to review, we appreciate your careful feedback, and the highlighting of the broad scope of the experiments. We'll respond to the critical feedback below.
>
> > ...hard to interpret from Figure 2
>
> Thanks for this point. We've added dotted lines at the onset and end of grokking to make it easier to see the relationship to the rank. One can see that the divergence singular value plots occurs in the same place. Hopefully this helps!
>
> > ...remaining observations are relatively well known, both from theoretical and empirical points of view...
>
> We discuss this context in detail in the related work section. The main point is that we believe the scope of prior work is quite limited to deep linear settings in many cases, with some experiments on larger neural networks, but far off from the scope of this work. Is there something in that discussion that was lacking, or some particular paper that we may have missed?
>
> Also, the results on linear mode connectivity, random labels, and lottery tickets are not covered at all by prior work to the best of our knowledge.

---

> ### Author Response · Authors · 2025-06-12
> **Thank you again for your review**
>
> Thank you again for your review. As recommendations will be due soon, please let us know if there are additional concerns after our rebuttal. Thanks again for your time.

---

### Decision · Action_Editor_Hkrn · 2025-06-27

**Recommendation:** Reject

**Additional Comments:**

This paper undertakes an extensive empirical investigation proposing that weight matrix rank minimization represents a universal phenomenon that can explain various behaviors in deep networks, such as grokking and generalization. The primary focus of discussion was on whether the evidence was sufficient, with one key reviewer consistently arguing that the claims lacked convincing support. Although the authors incorporated new experiments and improved figure clarity, the fundamental issues regarding insufficient empirical rigor within individual experimental settings and unclear data presentations remained unaddressed. As such, the paper is not suitable for publication at this time, although the authors are encouraged to address the reviewer concerns and consider resubmitting a major revision.

**Audience:**

Yes

**Audience Explanation:**

Yes. The paper will likely be of interest to members of the community due to its ambitious scope, connecting the spectral dynamics of weights to a wide range of well-known phenomena including grokking, generalization, and the lottery ticket hypothesis. However, this interest is qualified by the work's limited novelty and the previously noted shortcomings in its empirical evidence.

**Claims And Evidence:**

No

**Claims Explanation:**

No. While the submission's large breadth of experiments across diverse tasks is a notable strength, its claims are ultimately not supported by convincing evidence because this breadth comes at the cost of sufficient empirical depth in any single setting. Furthermore, the reviewers highlighted issues with the presentation of results, such as plots with inconsistent color scales and averaging across varied seeds, which obscure some evidence and make it difficult to verify the central claims.

**Resubmission Of Major Revision:**

The authors may consider submitting a major revision at a later time.